# The phytopathogenic fungus *Sclerotinia sclerotiorum* detoxifies plant glucosinolate hydrolysis products via an isothiocyanate hydrolase

Jingyuan Chen [1], Chhana Ullah [1], Michael Reichelt [1], Franziska Beran [2], Zhi-Ling Yang[2], Jonathan Gershenzon [1], Almuth Hammerbacher [3✉] & Daniel G. Vassão [1✉]

Brassicales plants produce glucosinolates and myrosinases that generate toxic iso-thiocyanates conferring broad resistance against pathogens and herbivorous insects. Nevertheless, some cosmopolitan fungal pathogens, such as the necrotrophic white mold *Sclerotinia sclerotiorum*, are able to infect many plant hosts including glucosinolate producers. Here, we show that *S. sclerotiorum* infection activates the glucosinolate-myrosinase system, and isothiocyanates contribute to resistance against this fungus. *S. sclerotiorum* metabolizes isothiocyanates via two independent pathways: conjugation to glutathione and, more effec-tively, hydrolysis to amines. The latter pathway features an isothiocyanate hydrolase that is homologous to a previously characterized bacterial enzyme, and converts isothiocyanate into products that are not toxic to the fungus. The isothiocyanate hydrolase promotes fungal growth in the presence of the toxins, and contributes to the virulence of *S. sclerotiorum* on glucosinolate-producing plants.

[1] Department of Biochemistry, Max Planck Institute for Chemical Ecology, 07745 Jena, Germany. [2] Research Group Sequestration and Detoxification in Insects, Max Planck Institute for Chemical Ecology, 07745 Jena, Germany. [3] Department of Zoology and Entomology, Forestry and Agricultural Biotechnology Institute, University of Pretoria, Pretoria 0028, South Africa. ✉email: almuth.hammerbacher@fabi.up.ac.za; vassao@ice.mpg.de

Chemical defenses are used by organisms of all kingdoms to fend off their natural enemies[1,2], but the deployment of such defenses may negatively impact the producing organism. The production of non-poisonous pro-toxins that can be activated when needed is one strategy used by many plants to hinder the attacks of pathogens and herbivores, while avoiding auto-toxicity[2–4]. For example, plants of the order Brassicales produce amino acid-derived glucosinolates (GLs) which are activated by β-thioglucoside glucohydrolase enzymes (myrosinases) upon tissue damage to produce toxic isothiocyanates (ITCs) and nitriles[5]. This so-called "mustard oil bomb" provides defense against attack by many insect herbivores and some pathogens[6–8]. However, Brassica crops like rapeseed, cabbages, and broccoli still suffer severe attacks from particular insects and microbial pathogens[9–12].

The defensive roles of plant GLs have been studied more frequently in connection with herbivores than pathogens. Yet the Arabidopsis *pen2* mutant plants, which are deficient in their activation of indolic GLs, were found to be more susceptible to the powdery mildew fungi *Blumeria graminis* and *Erysiphe pisi*, suggesting that indolic GLs are a broad-spectrum antifungal defense[7,13]. Similarly, the adapted necrotroph *Botrytis cinerea* showed enhanced virulence on an Arabidopsis mutant line lacking indolic GLs[14]. On the other hand, aliphatic GLs also contribute to plant resistance against specific pathogens. While most of the work on GLs in pathogen resistance has involved the tryptophan-derived indolic GLs, an Arabidopsis mutant impaired in the production of its most abundant methionine-derived aliphatic GL, 4-methylsulfinylbutyl GL (4MSOB-GL), was found to be more susceptible to the fungal wilt pathogen *Fusarium oxysporum*[3].

The abilities of some herbivores and pathogens to colonize plants of the order Brassicales have been attributed to their metabolism of GLs and ITCs. Some specialized insect herbivores can metabolize GLs in a way that prevents the formation of toxic ITCs[15,16]. However, most herbivores do not block GL activation, but instead use the general glutathione (GSH)-dependent mercapturic acid pathway to conjugate GL hydrolysis products and thereby deactivate them[17]. Interestingly, bacteria have also been reported to detoxify ITC hydrolysis products, but in a manner that is remarkably distinct from those in herbivores, by directly degrading ITCs into carbonyl sulfide and their corresponding amines using metallo-β-lactamase (MBL) enzymes named ITC hydrolases (ITCases)[18,19]. The ITCase-encoding gene *SaxA* ("survival in Arabidopsis extracts") was first reported in virulent *Pseudomonas syringae* and shown to be required for bacterial resistance to ITCs[18]. Subsequently, *SaxA* was found in some gut-associated bacteria from herbivores feeding on Brassicales plants, where it was proposed to facilitate tolerance of host larvae toward dietary ITCs[20,21]. Recently, an amine and its corresponding acetylated derivative (acetamide) were detected as minor products of ITC metabolism in the flea beetle *Psylliodes chrysocephala*[22], but it is not known whether this conversion was performed by the insect itself or by its associated microbiota. Despite our knowledge on GL metabolism in insects and bacteria, we know very little about fungal metabolism of GLs and how pathogenic fungi colonizing Brassicaceae plants cope with GLs and their toxic hydrolysis products.

*Sclerotinia sclerotiorum* (Lib.) de Bary is one of the most destructive and geographically widely distributed fungal pathogens of plants, causing white mold disease in over 400 plant species all over the world, including in important GL-producing crucifer crop plants[10]. A previous study showed that aliphatic ITCs inhibit the growth of *S. sclerotiorum* in vitro, suggesting a potential defensive role of aliphatic GLs in crucifer plants against this fungal pathogen[23], while other studies described the selective metabolism of several non-GL chemical defenses of the Brassicaceae.

Here, we study the role of GLs in the interaction of plants with *S. sclerotiorum* in more detail and investigate the metabolic mechanisms that allow the fungus to circumvent GLs during plant infection. Our results support the defensive role of methionine-derived, aliphatic GLs against this fungus, and show that the GL-myrosinase system is activated to form toxic ITCs during infection by *S. sclerotiorum*. However, this fungus efficiently degrades ITCs during plant infection, chiefly via enzymatic hydrolysis. We identify and characterize the ITCase responsible for this reaction, and show that this degradation pathway effectively reduces the toxicity of ITCs. Finally, phenotypic analysis of *S. sclerotiorum* ITCase-deficient mutants demonstrates that this ITC degradation pathway is critical for fungal colonization of GL-producing host plants.

## Results

**ITC-deficient plants are more susceptible to *S. sclerotiorum*.** To investigate the significance of aliphatic GLs and their activation products in plant defense against *S. sclerotiorum*, Arabidopsis wild-type (Col-0 WT) and mutant plants (*tgg1/tgg2* plants defective in leaf myrosinases and *myb28/myb29* plants deficient in aliphatic GL biosynthesis[24,25]) were separately infected with the fungus, and lesion sizes on individual leaves and relative fungal colonization were compared 24 h post inoculation (hpi). *S. sclerotiorum* caused around 50% larger leaf lesions and more severe tissue decay symptoms on *tgg1/tgg2* and *myb28/myb29* leaves than on WT leaves (Fig. 1a). Consistent with these visual symptoms, *S. sclerotiorum* colonization (quantified as the normalized expression of a fungal *Histone* housekeeping gene) was higher on both *tgg1/tgg2* and *myb28/myb29* mutant lines compared with WT plants (Fig. 1b). The *tgg1/tgg2* plants have normal GL contents but are defective in GL activation, causing substantial reduction in the formation of ITCs and other hydrolysis products. Meanwhile, *myb28/myb29* plants are deficient in both aliphatic GLs and their resulting ITCs. Since both mutants showed similar susceptibility to the fungus, the aliphatic ITCs (and not the GLs themselves) appear to play a defensive role against *S. sclerotiorum* infection.

**<em>S. sclerotiorum</em> reduces ITC levels in leaves and in culture.** To determine whether levels of aliphatic GLs and ITCs change during fungal infection, 4MSOB-GL (the most abundant GL in *A. thaliana* Col-0 leaves) and its hydrolysis product 4MSOB-ITC were quantified in *S. sclerotiorum*- and mock-inoculated Arabidopsis leaves over a time course. There was no significant difference in 4MSOB-GL content between mock-inoculated and fungus-inoculated plants, suggesting that aliphatic GL production was not induced by fungal inoculation (Supplementary Fig. 1a). Interestingly, the levels of 4MSOB-ITC increased several-fold in fungus-infected leaves compared with mock-treated plants (Fig. 2a). The contents of 4MSOB-ITC peaked in infected Arabidopsis leaves at 24 hpi, returning to similar levels as in mock-inoculated leaves after 48–72 hpi. The rapid accumulation of 4MSOB-ITC upon fungal infection suggests that the GL-myrosinase system was activated by fungal colonization already during the initial stages of infection, while the reduced ITC levels in the later period could result from fungal degradation of ITCs.

To explore how the fungus modifies these compounds, cultures of *S. sclerotiorum* were incubated with 4MSOB-GL and 4MSOB-ITC separately, and samples from the media were analyzed by liquid chromatography tandem mass spectrometry (LC–MS/MS). Consistent with in vivo quantification in the plant, the

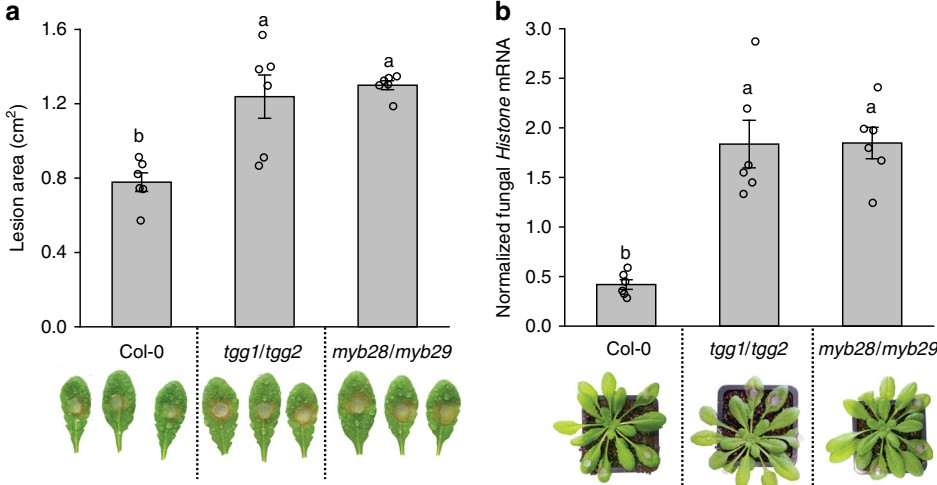

**Fig. 1 *A. thaliana* aliphatic ITC- and GL-deficient mutants are hyper-susceptible to *S. sclerotiorum*. a** Comparison of lesion areas caused by *S. sclerotiorum* 24 h post inoculation (hpi) on leaves of *A. thaliana* Col-0 wild-type, *tgg1/tgg2* and *myb28/myb29* mutants. The *tgg1/tgg2* line is a myrosinase-defective mutant, and the *myb28/myb29* line is deficient in aliphatic GL biosynthesis. **b** Relative quantification of *S. sclerotiorum* growth on *A. thaliana* lines for 24 h as quantified by qRT-PCR. The *S. sclerotiorum Histone* mRNA was normalized by the *A. thaliana ACTIN2* mRNA to quantify relative fungal colonization. Data represent mean ± SEM ($n = 6$ inoculated leaves and plants, respectively; inoculation with detached leaves was repeated once) and were analyzed by one-way ANOVA ($p < 0.001$) followed by Tukey's post-hoc test. Different letters above the bars indicate significant differences at $p < 0.05$. Source data are provided as a Source data file.

concentration of 4MSOB-GL was stable in the medium co-incubated with *S. sclerotiorum* over 48 h (Supplementary Fig. 1b), whereas the levels of 4MSOB-ITC decreased linearly within 12 h (Fig. 2b), supporting the fungal degradation of this compound. Meanwhile, various fungal transformation products including the glutathione (4MSOB-GSH), cysteinylglycine (4MSOB-CG), cysteine (4MSOB-CYS) and *N*-acetylcysteine (4MSOB-NAC) conjugates of 4MSOB-ITC, as well as 4-methylsulfinylbutylamine (4MSOB-amine) and 4-methylsulfinylbutylacetamide (4MSOB-acetamide) were detected (Fig. 2c, d). It is noteworthy that the total amount of the mercapturic acid conjugation pathway products (4MSOB-GSH, 4MSOB-CG, 4MSOB-CYS and 4MSOB-NAC) was ~100-fold lower than the hydrolytic products (4MSOB-amine and 4MSOB-acetamide), and amounts of the former did not increase further after 4 h of incubation (Fig. 2c). Conversely, the major hydrolysis product 4MSOB-amine increased significantly from 4 to 12 h (Fig. 2d), and was slowly further acetylated to form 4MSOB-acetamide. Therefore, we propose that *S. sclerotiorum* reduces the concentration of 4MSOB-ITC through two ITC transformation pathways, the mercapturic acid conjugation pathway and a hydrolytic degradation pathway, with the latter being more efficient (Fig. 2e).

**ITCs induce expression of a *S. sclerotiorum* ITCase-encoding gene.** In order to identify the proteins and corresponding genes responsible for ITC degradation in *S. sclerotiorum*, we searched for *S. sclerotiorum* protein homologs of a known ITCase, SaxA from *P. syringae*[18]. The ITCases belong to the MBL enzyme superfamily. Five MBL proteins of *S. sclerotiorum* (SS1G_12040, SS1G_11053, SS1G_01079, SS1G_14439 and SS1G_12145) were selected for further study based on their conserved domain structures and sizes[21]. The amino acid sequence identity among these five MBL proteins was lower than 15%, and a phylogenetic analysis together with MBL proteins from other closely related fungal species (and characterized ITCases from bacteria as outgroups) showed that each *S. sclerotiorum* candidate formed a separate cluster together with their putative homologs from related fungal species. With only 20% identity to *P. syringae* SaxA,

SS1G_12040 is nevertheless the *S. sclerotiorum* MBL protein closest to the previously characterized SaxA proteins (Fig. 3a).

We next used RT-qPCR to investigate whether the expression of the *S. sclerotiorum* ITCase candidate genes changed upon fungal exposure to an ITC-amended medium. The transcript level of *Ss12040* was 175-fold higher after exposure to 4MSOB-ITC (Fig. 3b), whereas expression of the other four candidate genes did not change (Supplementary Fig. 2). In order to study the expression of these genes under more natural conditions, we also analyzed their transcript abundances after inoculation of Arabidopsis Col-0 WT, *tgg1/tgg2* and *myb28/myb29* mutants, where the latter two lines do not produce aliphatic ITCs either before or after infection. Consistent with the in vitro expression of the five candidate genes, only *Ss12040* showed significantly higher expression (~14-fold) in *S. sclerotiorum* infecting the WT (Col-0) plants compared with *S. sclerotiorum* infecting the *tgg1/tgg2* and *myb28/myb29* mutants (Fig. 3c, Supplementary Fig. 3). *Ss12040* expression was also induced by 8-methylsulfinyloctyl ITC (8MSOO-ITC), which was suggested to be the most toxic aliphatic ITC in Arabidopsis against phytopathogens[23], and by an aromatic ITC, 2-phenylethyl ITC (2PE-ITC), an important benzenic ITC formed in the roots of several Brassicaceae plants[26] (Supplementary Fig. 4). Interestingly, *Ss12040* expression did not respond to exposure to the nitrile derived from 2PE-GL, 3-phenylpropanenitrile (2PE-CN) (Supplementary Fig. 4b).

**SsSaxA (Ss12040) degrades aliphatic and aromatic ITCs.** To determine the activity of the protein encoded by the *SaxA* candidate gene *Ss12040*, its 807 bp open reading frame was cloned and expressed in *Escherichia coli*. The enzyme activity was measured in vitro with 4MSOB-ITC, 8MSOO-ITC, and 2PE-ITC. Ss12040 was able to hydrolyze all ITCs offered in vitro to form the corresponding amine (Fig. 4, Supplementary Fig. 5). We named this protein SsSaxA, after its previously characterized bacterial homologs[18]. The SsSaxA enzyme followed classical Michaelis–Menten kinetic behavior with both 4MSOB-ITC and 2PE-ITC (Fig. 4), with $K_M$ values of 415 and 508 μM, respectively. Moreover, $V_{max}$ and $K_{cat}$ of SsSaxA for 4MSOB-ITC were 13.1 μmol min$^{-1}$ and 2.79 s$^{-1}$, and for 2PE-ITC 8.2 μmol min$^{-1}$ and 0.70 s$^{-1}$, suggesting

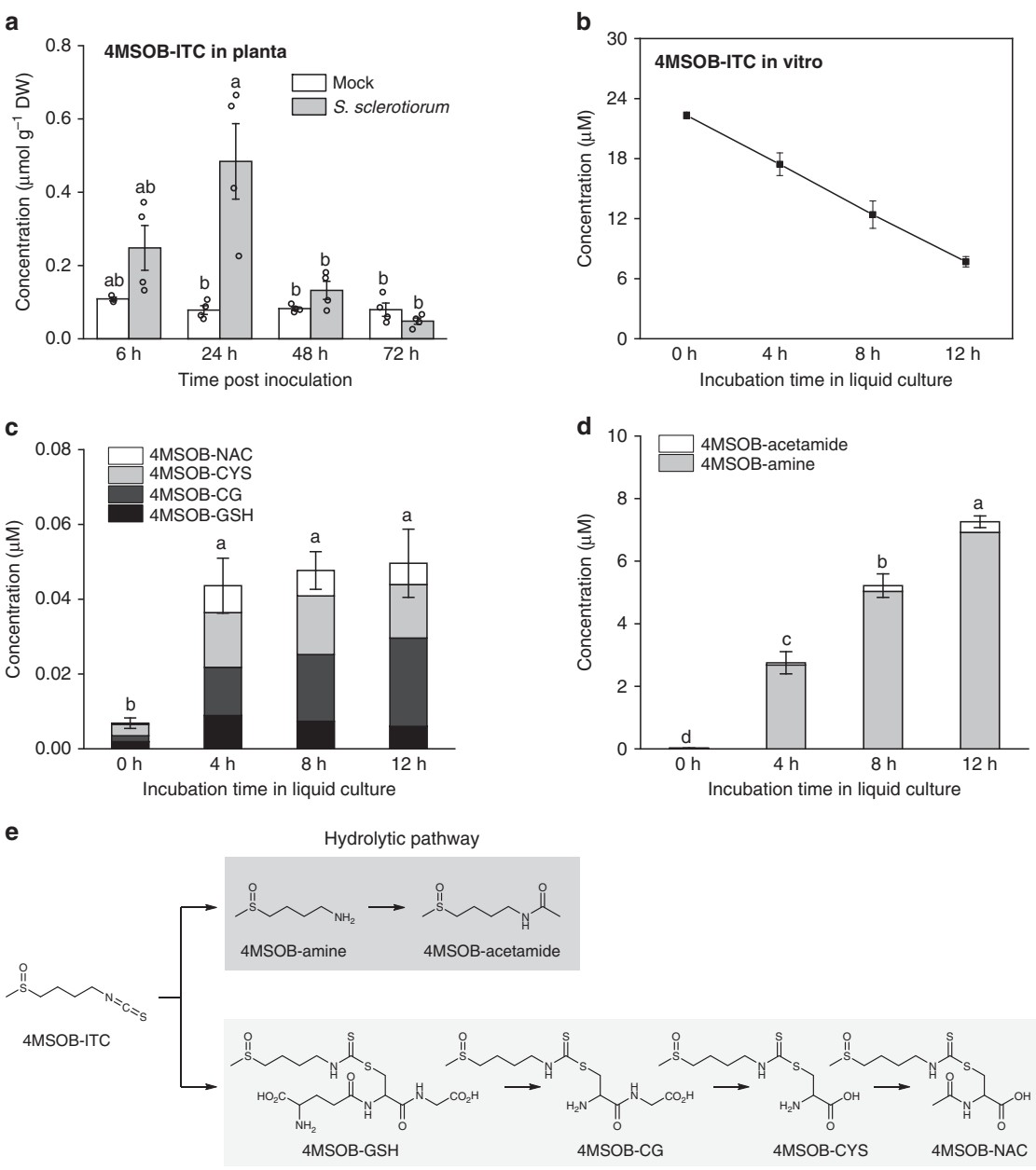

**Fig. 2 4MSOB-ITC is degraded by the fungus S. sclerotiorum. a** Quantification of 4MSOB-ITC in *A. thaliana* plants with and without *S. sclerotiorum*. Data represent mean ± SEM ($n = 4$ inoculated plants) and were analyzed by a Kruskal–Wallis rank sum test ($p < 0.01$) followed by a Games–Howell post-hoc test. Different letters above the bars indicate significant differences at $p < 0.05$. **b** Quantification of 4MSOB-ITC in fungal cultures during a time course. Twenty-five micromolar 4MSOB-ITC was used for each fungal culture. Data were analyzed by a linear regression ($R^2 = 0.94$, $p < 0.001$). Quantification of **c** 4MSOB-ITC mercapturic acid pathway conjugates, and **d** 4MSOB-ITC hydrolytic degradation products 4MSOB-amine and 4MSOB-acetamide, in the fungus-inoculated liquid medium supplemented with 4MSOB-ITC. Data represent mean ± SEM ($n = 3$ independent fungal cultures) and were analyzed by one-way ANOVA ($p < 0.001$) followed by Tukey's post-hoc test. Different letters above the bars indicate significant differences at $p < 0.05$. **e** Proposed pathways for the degradation of 4MSOB-ITC by *S. sclerotiorum*. 4MSOB-ITC, 4-methylsulfinylbutyl isothiocyanate; 4MSOB-GSH, 4MSOB-ITC glutathione conjugate; 4MSOB-CG, 4MSOB-ITC cysteinylglycine conjugate; 4MSOB-CYS, 4MSOB-ITC cysteine conjugate; 4MSOB-NAC, 4MSOB-ITC N-acetylcysteine conjugate; 4MSOB-amine, 4-methylsulfinylbutylamine; 4MSOB-acetamide, 4-methylsulfinylbutylacetamide. Source data are provided as a Source data file.

that SsSaxA accepts both classes of ITCs as substrates with slightly higher efficiency for the aliphatic ITC. In addition, *S. sclerotiorum* degraded 8MSOO-ITC more rapidly than 4MSOB-ITC (Supplementary Fig. 6).

**SsSaxA-mediated hydrolysis detoxifies ITCs in S. sclerotiorum.** To evaluate the biological significance of ITC hydrolysis for

*S. sclerotiorum*, we compared the toxic effects of 2PE-ITC on fungal growth with the effects of its hydrolysis products. Due to the commercial availability of 2-phenylethylamine (2PE-amine) and 2-phenylethylacetamide (2PE-acetamide), 2PE-ITC was used instead of 4MSOB-ITC. Radial mycelial growth was measured on potato dextrose agar (PDA) plates containing a series of concentrations of these compounds dissolved in dimethyl sulfoxide

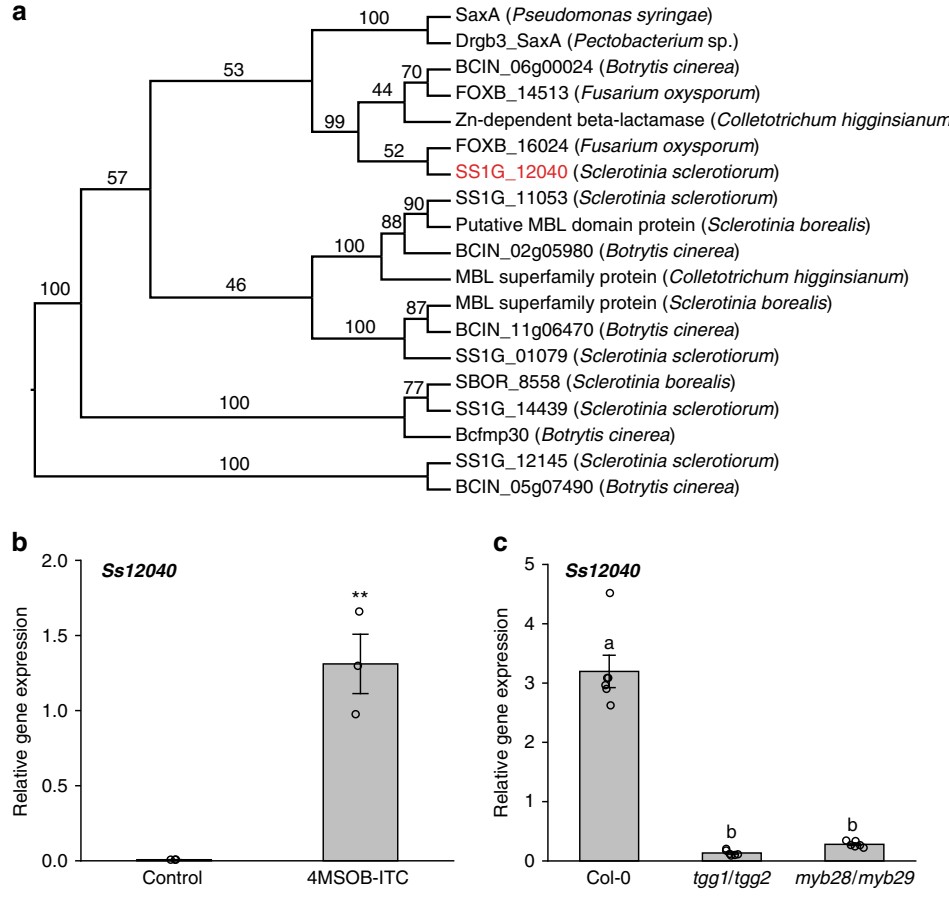

**Fig. 3 Ss12040 is a SaxA candidate gene in S. sclerotiorum. a** Maximum likelihood tree of fungal metallo-β-lactamase-like enzymes with similarity to known bacterial ITCases. Amino acid sequences were aligned with MEGA, and phylogenetic trees were constructed with PhyML using 1000 bootstraps. Numbers on each branch in the ML tree signify bootstrap values. **b** Induction of *SaxA* candidate gene *Ss12040* expression by 4MSOB-ITC in vitro. The fungal culture medium was supplemented with 4MSOB-ITC (25 μM), with only ethanol being added to the control medium. Gene expression was determined 30 min post inoculation. Data represent mean ± SEM ($n = 3$ independent fungal cultures; in vitro induction of *Ss12040* by 4MSOB-ITC was repeated once) and were analyzed by a two-tailed Student's *t* test (**$p < 0.01$). **c** Expression of *SaxA* candidate gene *Ss12040* 24 hpi of different *A. thaliana* lines. Data represent mean ± SEM ($n = 6$ independent inoculated plants) and were analyzed by a Kruskal–Wallis rank sum test ($p < 0.001$) followed by a Games–Howell post-hoc test. Different letters above the bars indicate significant differences at $p < 0.05$. 4MSOB-ITC, 4-methylsulfinylbutyl isothiocyanate. Source data are provided as a Source data file.

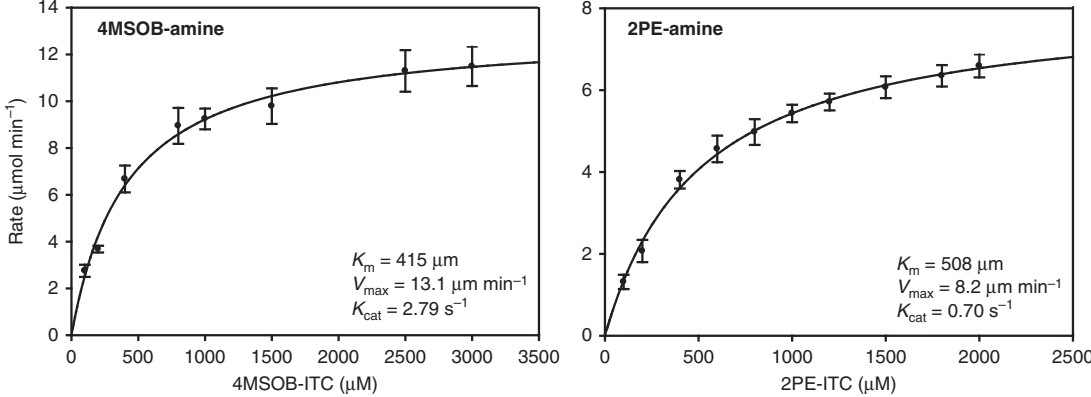

**Fig. 4 Enzyme kinetic analyses of heterologously expressed *SsSaxA* with 4MSOB-ITC and 2PE-ITC.** 4MSOB-amine was analyzed as its FMOC (fluorenylmethoxycarbonyl chloride) derivative to avoid signal quenching by the buffer. Products in each reaction were measured 5 min after the enzyme was added. Data represent mean ± SEM ($n = 3$ independent reactions). 4MSOB-ITC, 4-methylsulfinylbutyl isothiocyanate; 4MSOB-amine, 4-methylsulfinylbutylamine; 2PE-amine, 2-phenylethylamine; 2PE-ITC, 2-phenylethyl isothiocyanate. Source data are provided as a Source data file.

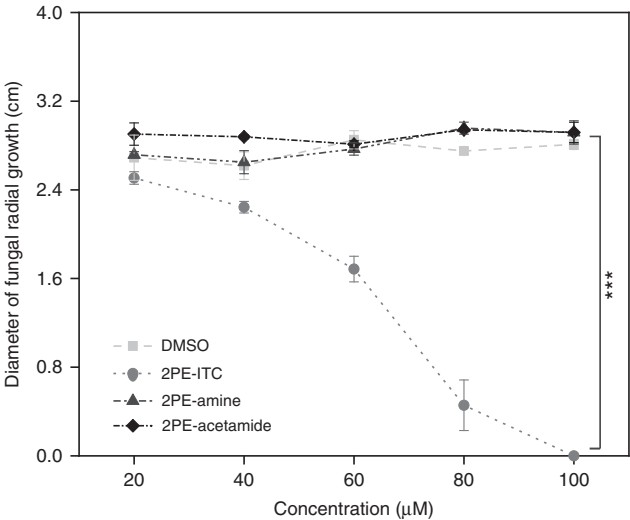

**Fig. 5 2PE-ITC but not its metabolites 2PE-amine and 2PE-acetamide reduces *S. sclerotiorum* growth.** *S. sclerotiorum* was grown on PDA plates amended with different concentrations of these three metabolites and the diameter of fungal radial growth was measured 24 h after co-incubation with these compounds. Data represent mean ± SEM (*n* = 3 independent fungal cultures; growth assay was repeated once) and were analyzed by fitting a linear model using ANCOVA (2PE-ITC/DMSO, 2PE-ITC/2PE-Amine and 2PE-ITC/2PE-Acetamide, ***$p < 0.001$). 2PE-ITC, 2-phenylethyl isothiocyanate; 2PE-Amine, 2-phenylethylamine; 2PE-Acetamide, 2-phenylethylacetamide. Source data are provided as a Source data file.

(DMSO), with DMSO alone applied as a control. *S. sclerotiorum* growth inhibition by 2PE-ITC was dose-dependent and already detectable around 40 µM (Fig. 5). However, fungal growth inhibition in media amended with DMSO, 2PE-amine or 2PE-acetamide was not detected up to 100 µM, showing that these compounds were not toxic to the fungus over this range (Fig. 5). Therefore, SsSaxA converts the toxic 2PE-ITC to non-toxic products indicating that hydrolysis of ITCs in *S. sclerotiorum* is a true enzymatic detoxification process (Fig. 5). Interestingly, the expression of *SsSaxA* did not correlate well with 2PE-ITC concentration, although this compound's toxicity was dose-dependent (Supplementary Fig. 4b). The significantly lower induction of *SsSaxA* expression under high 2PE-ITC concentrations might also attributed to the severe inhibition of fungal growth by 2PE-ITC at concentrations higher than 40 µM.

**SsSaxA knockout mutants are sensitive to ITCs in culture.** To experimentally manipulate the in vivo ITC degradation activity of SsSaxA in *S. sclerotiorum*, we generated ITCase knockout mutants by PEG-mediated transformation and homologous recombination. Successful deletion of the *SsSaxA* gene was confirmed by amplification of the resistance marker gene (*Hyg*) and the recombination cassette in genomic DNA of Δ*SsSaxA*-(*1–3*) mutants (Supplementary Fig. 7a). Moreover, semi-quantitative PCR of *SsSaxA* cDNA from both WT and mutant fungi exposed to 2PE-ITC (DMSO as control) showed that *SsSaxA* was only expressed in WT mycelia treated with solvent or 2PE-ITC (Supplementary Fig. 7b).

To determine whether the deletion of the *SsSaxA* gene affected ITC metabolism in *S. sclerotiorum*, WT and Δ*SsSaxA* mutants were cultured in liquid medium amended with 4MSOB-ITC, and the ITC degradation rates were compared through a time course. The WT fungus degraded 4MSOB-ITC significantly faster than the Δ*SsSaxA* mutants (Fig. 6a; Supplementary Fig. 8a), and the concentration of 4MSOB-ITC remaining in mutant cultures after

12 h co-incubation was twice that of the WT cultures. In addition to the lower efficiency of 4MSOB-ITC degradation, we observed that the degradation products, 4MSOB-amine and -acetamide, only accumulated in WT fungal cultures, but not in Δ*SsSaxA* mutant cultures (Fig. 6b). Surprisingly, 4MSOB-ITC conjugates from the mercapturic acid pathway were significantly more abundant in *SsSaxA* deletion mutant cultures than in WT fungal cultures (Fig. 6c), suggesting that the conjugation pathway might partially compensate for the loss of the hydrolytic pathway in *S. sclerotiorum* Δ*SsSaxA*. A trade-off between the ITC hydrolysis and the conjugation pathways was also observed directly in mycelia of Δ*SsSaxA* mutants, which accumulated the 4MSOB-ITC substrate and its conjugates, but not the amine product (Supplementary Fig. 9). The lack of ITC degradation in Δ*SsSaxA* mutants was further confirmed by incubation with 8MSOO-ITC and 2PE-ITC, with the corresponding hydrolytic degradation products not formed by mutant cultures (Supplementary Fig. 10). These results showed that *SsSaxA* is responsible for ITC hydrolysis in *S. sclerotiorum*. Since the concentration of 4MSOB-ITC conjugates in mutant cultures was significantly lower than the concentration of hydrolytic degradation products in WT cultures, the conjugation pathway seems to be a less effective ITC detoxification system than the hydrolytic pathway.

To further investigate the importance of the hydrolytic pathway for fungal tolerance to ITCs, the growth of both WT and Δ*SsSaxA* fungi was measured after exposure to a series of concentrations of 4MSOB-ITC and 2PE-ITC dissolved in ethanol (EtOH) and DMSO, respectively. There was no significant difference in the growth rates of WT and Δ*SsSaxA* mutant fungi on the control plates amended with solvent only (Fig. 6d, e), indicating that *SaxA* is not essential for the growth and development of *S. sclerotiorum* under normal conditions. However, ITCase deficiency resulted in dramatically reduced growth in the presence of both 4MSOB-ITC and 2PE-ITC (Fig. 6d, e). Hence, *SsSaxA*-dependent ITC detoxification allows *S. sclerotiorum* to tolerate ITCs in the medium.

**SaxA deletion reduces fungal pathogenicity on ITC-defended plants.** The role of ITC detoxification during fungal infection of GL-containing plants was tested by quantifying the pathogenicity of both WT and Δ*SsSaxA* fungi on WT Arabidopsis Col-0, as well as Arabidopsis mutants producing altered levels of ITCs, and other cultivated Brassicaceae species. Deletion of ITCase significantly reduced the lesion areas and fungal colonization of the Δ*SsSaxA* mutants on ITC-forming Arabidopsis Col-0 compared with the WT fungus (Fig. 7). The reduced pathogenicity of Δ*SsSaxA* mutants was partially recovered when infecting leaves of the Arabidopsis mutants *tgg1/tgg2* and *myb28/myb29*, which do not produce aliphatic ITCs and GLs, respectively (Supplementary Fig. 11a, b). The Arabidopsis epithiospecifier protein (ESP) overexpression line *35S:ESP*[27], which produces nitriles instead of ITCs during GL hydrolysis, exhibited similar resistance when inoculated with WT fungus and two Δ*SsSaxA* mutants (Supplementary Fig. 11c), suggesting that reduced ITC formation restores the pathogenicity of the deletion mutants. To examine the role of fungal ITCase in the metabolism of indolic GL hydrolytic products, we compared the lesion areas caused by WT and Δ*SsSaxA* mutant fungi on the Arabidopsis *cyp79b2/b3* double knockout mutant, which has reduced levels of both indolic GLs and camalexin but retains normal levels of aliphatic GLs[28]. *Cyp79b2/b3* plants were more susceptible to infection by the WT fungus than the Δ*SsSaxA* mutants (Supplementary Fig. 11d) indicating that these indolic products do not fully dictate plant resistance to this fungal pathogen. Furthermore, pathogenicity assays were conducted on two cultivated Brassica plants, *Nasturtium*

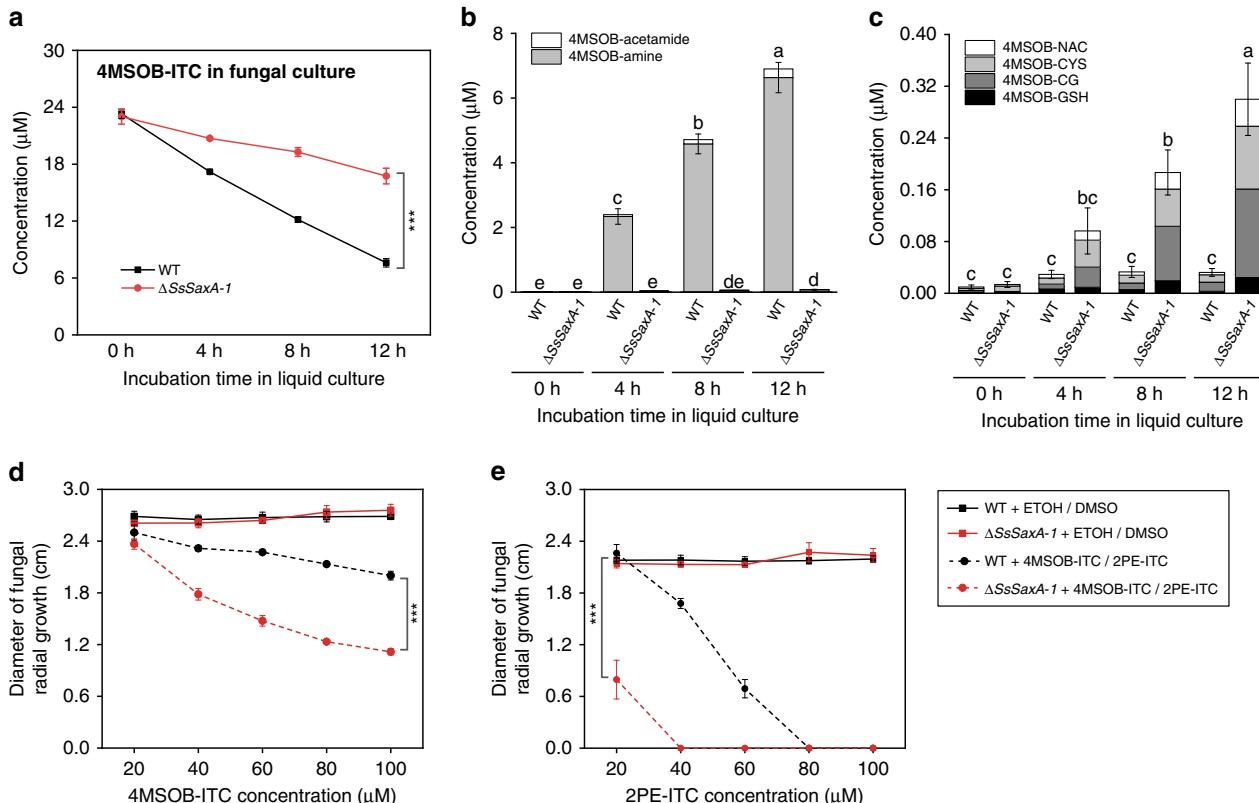

**Fig. 6 SsSaxA deletion reduces ITC degradation efficiency and growth in ITC-containing medium. a** Quantification of 4MSOB-ITC in both wild-type (WT) and ΔSsSaxA-1 cultures over a time course. Twenty-five micromolar 4MSOB-ITC was used for each culture. Data represent mean ± SEM ($n = 3$ independent fungal cultures) and were analyzed by one-way ANCOVA (***$p < 0.001$ for WT/ΔSsSaxA-1). Quantification of the degradation products **b** 4MSOB-amine and 4MSOB-acetamide and **c** 4MSOB-ITC conjugates in WT and ΔSsSaxA-1 cultures supplemented with 4MSOB-ITC. For degradation products, data were analyzed by a Kruskal–Wallis rank sum test ($p < 0.01$) followed by a Games–Howell post-hoc test; for conjugates, data were analyzed by one-way ANOVA ($p < 0.001$) followed by Tukey's post-hoc test. Different letters above the bars indicate significant differences at $p < 0.05$. Deletion of SsSaxA in S. sclerotiorum resulted in reduced growth in culture medium in the presence of both **d** 4MSOB-ITC and **e** 2PE-ITC. Diameter of fungal radial growth was measured 24 h after co-incubation with these compounds. Data represent mean ± SEM ($n = 3$ independent fungal culture; growth assays of WT and ΔSsSaxA mutants with ITCs were repeated once) and were analyzed by one-way ANCOVA (4MSOB-ITC or 2PE-ITC treatment, ***$p < 0.001$ for WT/ΔSsSaxA-1). 4MSOB-ITC, 4-methylsulfinylbutyl isothiocyanate; 4MSOB-GSH, 4MSOB-ITC glutathione conjugate; 4MSOB-CG, 4MSOB-ITC cysteinylglycine conjugate; 4MSOB-CYS, 4MSOB-ITC cysteine conjugate; 4MSOB-NAC, 4MSOB-ITC N-acetylcysteine conjugate; 4MSOB-amine, 4-methylsulfinylbutylamine; 4MSOB-acetamide, 4-methylsulfinylbutylacetamide; 2PE-ITC, 2-phenylethyl isothiocyanate. Source data are provided as a Source data file.

officinale and Sinapis alba, which contain 2-phenylethyl and 4-hydroxybenzyl GL as the major GLs, respectively[29,30]. After inoculation, ΔSsSaxA mutants showed dramatically reduced pathogenicity on both plants compared with the WT fungus (Supplementary Fig. 12). Overall, our results show that ITC detoxification via the hydrolysis pathway in S. sclerotiorum is important for fungal virulence on plants with an intact GL-myrosinase defense system.

## Discussion

In this study, we found that a major group of two-component defenses in plants, the GL-myrosinase system of the Brassicales, plays an important role in defense against fungal infection. The activation of GLs to toxic ITC hydrolysis products helps reduce the susceptibility of Arabidopsis to the necrotrophic pathogen S. sclerotiorum. However, we also discovered that this generalist fungus with an extremely broad host range possesses a detoxification mechanism that specifically circumvents the toxic effects of ITCs.

Our first indication that GLs have defensive properties against S. sclerotiorum came from our finding that the Arabidopsis myb28/myb29 mutant, which is deficient in aliphatic GLs, was

more susceptible to infection than plants with WT GL levels. Next we showed that the colonization of S. sclerotiorum on the tgg1/tgg2 mutant, which lacks the leaf myrosinases that activate GLs, was higher than on WT Col-0 plants, but similar to colonization on myb28/myb29. Thus, we inferred that aliphatic GLs contribute to plant defense against pathogenic fungi via their hydrolysis products (ITCs). The toxicity of ITCs is attributed to their electrophilic and lipophilic characteristics, which allow them to cross cellular membranes and react with specific intracellular targets[31]. For instance, ITCs were able to cause intracellular accumulation of reactive oxygen species in the necrotrophic fungal pathogen Alternaria brassicicola[32]. In our study, we observed that 4MSOB-ITC concentrations in inoculated plants were significantly elevated at 6 hpi and reached maximum levels at 24 hpi, which suggested the activation of the GL-myrosinase defense system in the initial stages of fungal infection. Nevertheless, the substantial decrease in 4MSOB-ITC content that occurred after 48 hpi in fungal inoculated-Arabidopsis leaves suggested that S. sclerotiorum might actively reduce the levels of these toxic GL hydrolysis products during the later stages of infection.

Previous studies have suggested ways in which fungi might resist plant-produced ITCs. The MAP kinase AbHog1 and the

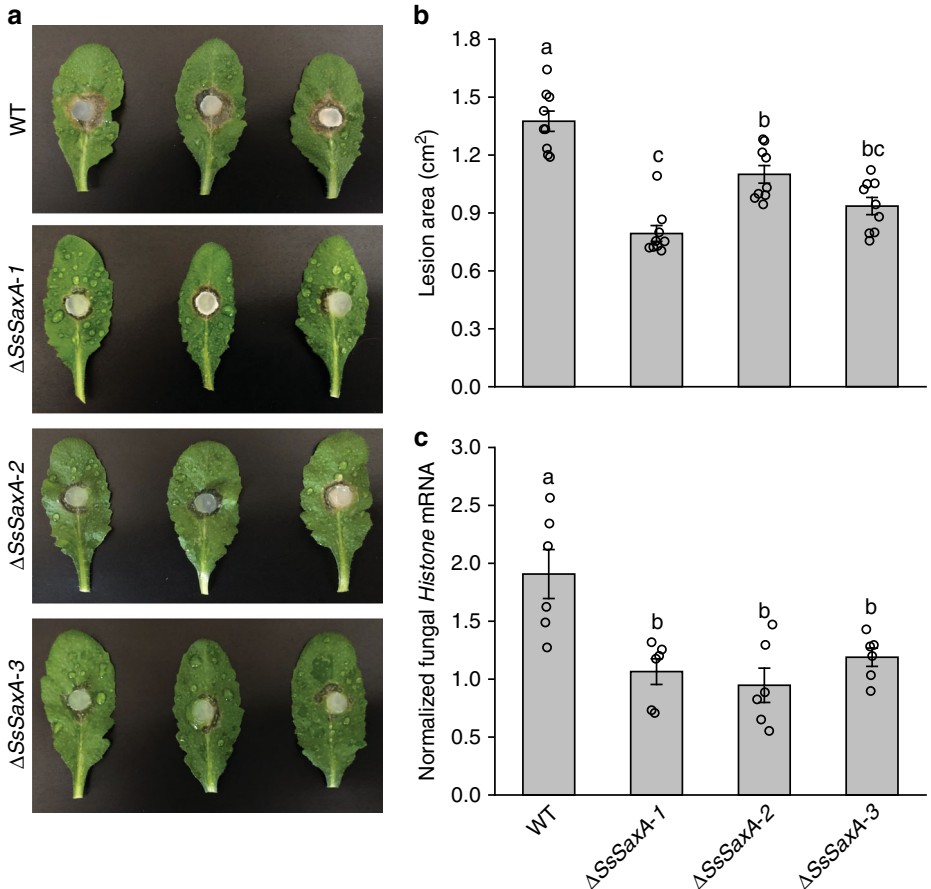

**Fig. 7 ΔSsSaxA mutants are significantly less virulent on *A. thaliana* Col-0 than the wild-type fungus. a** Representative images of *A. thaliana* Col-0 leaves infected by WT and Δ*SsSaxA* mutants. **b** Comparison of lesion area caused by WT and Δ*SsSaxA* mutants 24 hpi on leaves of *A. thaliana*. Data represent mean ± SEM (*n* = 9 inoculated leaves from separate plants). **c** Relative growth of WT and Δ*SsSaxA* mutants on *A. thaliana* Col-0 as determined by qRT-PCR. The fungal *Histone* mRNA was normalized to the *A. thaliana ACTIN2* mRNA to quantify the relative fungal colonization 24 hpi. Data represent mean ± SEM (*n* = 6 independent inoculated plants). Data were analyzed by one-way ANOVA (*p* < 0.001) followed by a Tukey's post-hoc test. Different letters above the bars indicate statistically significant differences at *p* < 0.05. Source data are provided as a Source data file.

transcription factor *Ab*AP1, which are important regulators of the response to oxidative stress, were found to be essential for *A. brassicicola* growth in the presence of ITCs[32]. Moreover, transcriptional analysis of the *A. brassicicola* response to the aliphatic allyl-ITC showed the induction of some membrane transporters as well as glutathione-*S*-transferases (GSTs)[33,34]. Recently, the major facilitator superfamily transporter *mfsG* was shown to mediate ITC efflux and therefore provide resistance to ITCs in *B. cinerea*[35].

Although these previous studies focused on the excretion of ITCs and mitigation of their toxic effects, our findings suggested metabolic transformation as a resistance mechanism, since *S. sclerotiorum* is able to deplete the levels of 4MSOB-ITC in liquid culture within 12 h, yielding both degradation products and conjugates. The conjugation of 4MSOB-ITC to glutathione to generate more polar products is common in insect herbivores and serves to avoid the toxic effects of ITCs[36–38]. In fact, the expression of several GST candidate genes was shown to be highly induced in *S. sclerotiorum* by allyl-ITC or mustard powder[39]. However, we showed that the conjugates formed from 4MSOB-ITC in the mercapturic acid pathway in *S. sclerotiorum* cultures appeared only at low levels 4 h after addition, and were not further elevated.

Instead, the hydrolysis of ITCs to their corresponding amines and acetamides seems to represent a more important metabolic process. The hydrolysis of ITCs to amines had been previously described in plant pathogenic bacteria, insect gut-associated bacteria and the flea beetle, *Psylliodes chrysocephala*[18,19,21,22], but not until now in fungi. Interestingly GL-producing plants themselves are also reported to generate GL-derived amines[7], albeit via a very different pathway using mercapturic acid pathway products instead of the ITC hydrolytic pathway described in microbes[22].

The two different pathways to metabolize ITCs in *S. sclerotiorum* might be important under different conditions. For example, the conjugation pathway might be utilized only for a rapid, early response to ITCs and the hydrolytic pathway after pathogen establishment on the host plant. ITC detoxification via GSH conjugation may be more costly for the fungus than hydrolysis as demonstrated for the insect herbivore *Spodoptera littoralis*, where conjugation resulted in reduced GSH, cysteine, and protein content and decreased larval growth[40].

The hydrolysis of ITCs by *S. sclerotiorum* is carried out by MBL-like enzymes that are widely distributed in pathogenic bacteria and previously reported to confer resistance to ITCs[21]. Encoded by the so-called *SaxA* genes, these proteins form their own subfamily within the well-studied MBL superfamily. In our search for a SaxA homolog in *S. sclerotiorum*, we considered five candidate MBL-like genes that encode enzymes with only low

amino acid sequence identity to each other. Among these potential candidates, only *Ss12040* was induced by treatment with different ITCs, or during colonization of WT Arabidopsis compared with the aliphatic ITC-deficient mutants (*myb28/myb29* and *tgg1/tgg2*). Homologs of the protein encoded by *Ss12040* (SsSaxA) are also present in other fungal pathogens infecting Brassicales and form a single well-supported cluster. Accordingly, these fungal pathogens on Brassicales plants might also have the ability to hydrolyze ITCs to amines. A potential SaxA homolog in a related species, *Sclerotinia borealis*, was not included in our phylogenetic analysis because of a premature stop codon at its amino acid position 112 (accession number: ESZ95787, 65% identity to SsSaxA). Some bacteria colonizing Brassicales plants possess several SaxA homologs, with at least one having a signal peptide[21]. However, the *S. sclerotiorum* SaxA does not appear to contain a signal peptide. Regardless of its subcellular location, the *S. sclerotiorum* SaxA possesses similar affinities toward both aliphatic and benzenic ITCs. A previous study showed that bacterial ITCases are active with multiple substrates rather than having specificity to a single substrate[21]. The ability to break down chemically different ITCs may be very important for *S. sclerotiorum* as it infects a range of Brassicales with a diversity of GLs. Interestingly, 8MSOO-ITC and 2PE-ITC were apparently more efficiently degraded by fungal cultures than 4MSOB-ITC, with the transformation of 8MSOO-ITC and 2PE-ITC to the corresponding amines accomplished within 4 h, whereas more than 12 h were needed to transform an equal concentration of 4MSOB-ITC. These inferred ITC degradation rates are consistent with their toxicities, as 8MSOO-ITC and 2PE-ITC are more toxic to this fungus than 4MSOB-ITC[23], but how the in vivo degradation of ITCs with varying side-chains is differentially controlled by this fungus still needs to be further studied.

By fungal gene deletion, we generated ITCase-deficient *S. sclerotiorum* mutants, ΔSsSaxA-(1–3). This deletion reduced the efficiency of ITC hydrolysis in the fungus and reduced its growth in the presence of ITCs. Notably, when the hydrolytic pathway was impaired, ITC conjugates from the mercapturic acid pathway accumulated to a much higher level in both medium and mycelium of ΔSsSaxA mutants than of the WT fungus, suggesting a compensatory function of each pathway for the other. Although the conjugation pathway was activated to a higher level in ΔSsSaxA compared with the WT fungus, the deletion mutant grew more slowly in the presence of ITCs both in culture and during infection of GL-containing plants, indicating that the conjugation route is not as efficient in metabolizing ITCs as hydrolysis. On the other hand, the difference of fungal virulence between WT and ΔSsSaxA fungi was dramatically reduced during infection of Arabidopsis lines deficient in aliphatic ITC formation (*tgg1/tgg2* and *myb28/myb29* knockout mutants and a *35S:ESP* over-expressing line) compared with infection of Col-0. These results further support both the importance of ITCs in resistance against phytopathogens, and the role of the degradation of ITCs in fungal adaptation to the plant GL-myrosinase defenses.

In conclusion, our results clearly demonstrate that formation of aliphatic ITCs through activation of the GL-myrosinase system is an important plant defense mechanism against the necrotrophic fungal pathogen *S. sclerotiorum*. At the same time, this pathogen metabolizes ITCs to amines by a route previously undescribed in fungi that makes up the dominant ITC degradation pathway in *S. sclerotiorum* and represents a genuine detoxification reaction. The detoxification of ITCs is essential for fungal tolerance to GLs, the characteristic chemical defenses of Brassicales plants, and thus contributes to *S. sclerotiorum* virulence on the important crop species of this order, such as cabbage, broccoli, cauliflower, mustard and rapeseed.

## Materials and methods

**Fungal strain maintenance and plant cultivation**. The *S. sclerotiorum* 1980 strain[41] and ΔSsSaxA mutants (based on 1980 background) were grown on PDA at 25 °C to obtain vegetative growth. Arabidopsis WT ecotype Columbia (Col-0), the double knockout mutants (*tgg1/tgg2*, *myb28/myb29*, and *cyp79b2/b3*) and the *ESP* transgenic line (*35S:ESP*) (all based on Col-0 background) were used in this study[24,25,27,28]. All seeds were germinated on Murashige and Skoog (MS) medium and 1-week-old seedlings were transferred to potting soil (Klasmann-Deilmann, Geeste, Germany). Arabidopsis plants were grown under short-day conditions (10 h light/14 h dark cycle at 21 °C, 60% humidity). *Nasturtium officinale* (Rieger-Hofmann, Blaufelden-Raboldshausen, Germany) and *Sinapis alba* (N. L. Chrestensen, Erfurt, Germany) were germinated from seed in potting soil and grown under long-day conditions (16 h light/8 h dark cycle at 24 °C, 75% relative humidity) for 4–5 weeks.

**Plant inoculation**. Agar plugs (0.5 cm diameter) were punched from the actively growing edge of the fungal PDA plates. For in vivo inoculations, five rosette leaves of 4–5-week-old Arabidopsis plants were inoculated by placing the agar plugs on the leaf surface. Arabidopsis inoculated with PDA agar plugs without fungus was used as a control. Inoculation experiments were performed under short-day conditions and each treatment had 4–6 biological replicates. Arabidopsis leaves were harvested at different time points after inoculation and immediately flash frozen in liquid nitrogen for RNA isolation or compound extraction.

**Assessment of fungal infection levels**. Leaves from different Arabidopsis lines, *N. officinale* and *S. alba* were placed on water-saturated filter paper in a Petri dish and inoculated with fungal agar plugs. Petri dishes were incubated at 21 °C for 24 h. Five to nine replicates were used for each genotype and a photo of each leaf was taken. A 2 × 2 cm reference square was included in each photo. The magic wand tool in Photoshop CS5 was used to select symptomatic regions and reference regions of each photo. The ratio of pixel values of the symptomatic regions relative to the 4 cm² reference region was used to calculate the lesion area.

For relative quantification of fungal colonization, the amount of normalized fungal mRNA was calculated by the ratio of fungal *Histone* gene expression to the Arabidopsis *ACTIN2* gene expression. Inoculated Arabidopsis leaves were harvested and ground under liquid nitrogen. Total RNA was isolated with the Stratec Plant RNA Mini Kit (Birkenfeld, Germany) and DNA was eliminated with DNase (Qiagen, Hilden, Germany). The cDNA was synthesized by Superscript II reverse transcriptase kit (Invitrogen, Carlsbad, CA, USA) and used as template for a two-step quantitative RT-PCR using Brilliant III Ultra-Fast SYBR Green QPCR Master Mix (Agilent Technologies) with the following cycling conditions: 95 °C for 5 min, 40 cycles at 95 °C for 15 s and extension at 60 °C for 30 s. The primers used for amplification of housekeeping genes were as follows: SsH3-forward, GGCTC GTACCAAGCAAACTG and SsH3-reverse, GAAGTCTTGGGCGATTTCAC for the *S. sclerotiorum Histone* gene; AtAct2-forward, CTTGCACCAAGCAGCAT GAA and AtAct2-reverse, CCGATCCAGACA CTGTACTTCCTT for the Arabidopsis *ACTIN2* gene[42,43]. Data were processed with Bio-Rad CFX Real-Time PCR Systems.

**Extraction of GLs and ITCs from Arabidopsis leaves**. Arabidopsis leaves were lyophilized in an Alpha 1–4 LD Plus freeze dryer (Martin Christ GmbH, Osterode, Germany) for 2 days and ground to a fine powder with zirconium oxide beads in a shaking ball mill. Approximately 20 mg dry plant tissue was extracted as in Burow et al.[27] with 1 ml 80% methanol at room temperature for 10 min and then centrifuged at $18,000 \times g$ for 10 min. The supernatant was applied to a sephadex DEAE A-25 column (Sigma-Aldrich), which was then washed with 0.02 M MES buffer (pH 5.2) and incubated overnight at room temperature with arylsulfatase (Sigma-Aldrich). Distilled water (500 µl) was used to elute the desulfo-GLs.

For ITC extraction, 20 mg dry plant tissue was extracted with 500 µl 50% (v v⁻¹) methanol in water at pH 3 and agitated for 10 min on a horizontal shaker at room temperature. Samples were then centrifuged at $21,000 \times g$, 4 °C for 15 min and supernatants were transferred to LC vials for subsequent analyses.

**Collection of fungal breakdown products**. *S. sclerotiorum* was cultured in potato dextrose broth and incubated at 25 °C, shaking at 150 rpm for 2 days. ITCs (4MSOB-ITC/8MSOO-ITC/2PE-ITC) or 4MSOB-GL were added to the cultures to a final concentration of 25 µM. Liquid medium (200 µl) was collected immediately from fungal cultures at 0 h (control) and then at different time points. Samples were mixed with equal amounts of methanol and transferred to LC vials for LC–MS/MS analysis.

**Phylogenetic analysis of putative ITCases**. Searching in the NCBI protein database with the keyword *S. sclerotiorum* MBL, nine putative MBL proteins were found. Four of them had amino acid sequences much longer than those previously reported for ITCases and were thus removed from further analysis. The other five amino acid sequences and their homologs in selected close relatives, including *Botrytis cinerea*, *F. oxysporum*, *Collectotrichum higginsianum* and *S. borealis* (obtained by searching with BLAST (NCBI) in the RefSeq database), were used to construct the tree. Characterized ITCases (SaxA) from *P. syringae* and

*Pectobacterium* sp. were also included. The 19 amino acid sequences were aligned using MUSCLE in MEGA 5.05. The aligned sequences were used for best model selection after manual adjustment in MEGA. The maximum likelihood tree was constructed with the aligned sequences using the software PhyML v. 3.0[44]. The WAG + Gamma amino acid substitution model and 1000 non-parametric bootstrap analysis were performed. The tree was then visualized using FigTree v1.4.2 and Adobe Illustrator CS5. Accession numbers of all protein sequences used in this phylogenetic analysis are provided at the end of the "Methods" section.

**Quantitative real-time PCR of ITCase candidate genes**. *S. sclerotiorum* was incubated in liquid medium with 25 μM 4MSOB-ITC (dissolved in ethanol) for 30 min. An equal amount of EtOH was added to the fungal cultures used as controls. RNA isolation and reverse transcription-quantitative PCR (RT-qPCR) were performed using the program described for assessment of fungal infection levels. Primers for *SsSaxA* candidate genes were as follows: qRT-*Ss12040*-forward, CGTCAACTTCCTTACCGACCC and qRT-*Ss12040*-reverse, GCTGGCGACCAT AATCATCC; qRT-*Ss11053*-forward, TCTCCGTGCGGTTTATTCG and qRT-*Ss11053*-reverse, TGCTCCATGTCCTGGATATGC; qRT-*Ss01079*-forward, ACTC TGGGAAACGGGATGTG and qRT-*Ss01079*-reverse, ACCAGAATGCCGACCT CAAC; qRT-*Ss11439*-forward, AACCTTATACGCCGCCACC and qRT-*Ss11439*-reverse, ATTCGCCGGTTAGTTTCGC; qRT-*Ss12145*-forward, CAAGACACCG CTTGGAAATG and qRT-*Ss12145*-reverse, GGTCAAGTCGATTCAGCCAAC. Transcripts of these genes were normalized against the transcripts of the *S. sclerotiorum Histone* gene.

The inoculated Arabidopsis Col-0 WT and aliphatic ITC-deficient mutants (*tgg1/tgg2* and *myb28/myb29*) were harvested at 24 hpi for RNA isolation. The fungal-plant RNA samples were also used for RT-qPCR of *SsSaxA* candidate genes. The induction of the expression of the *SsSaxA/Ss12040* gene was confirmed by application of 25 μM 8MSOO-ITC (dissolved in EtOH) in fungal medium. Solvent only was supplied as control. To evaluate the correlation between ITC concentrations and *SsSaxA* gene expression, 100 mM 2PE-ITC in DMSO stock was diluted in liquid culture to yield a series of final concentrations (20, 40, 60, 80 and 100 μM), and amounts of DMSO equal to 100 μM 2PE-ITC were added to the control fungal cultures. Meanwhile, 100 μM 2PE-CN was also included to study whether expression of the *SsSaxA* gene can respond to the corresponding nitrile. After 30 min co-incubation with ITCs/solvent/2PE-CN, mycelium was harvested for RNA isolation and RT-qPCR of *SsSaxA* gene using the methods described above for assessment of fungal infection levels.

**Heterologous expression and characterization of SsSaxA (ITCase)**. The open reading frame of *SsSaxA* was amplified from *S. sclerotiorum* cDNA with the primers ORF-*SsSaxA*-forward: ATGTCGACTTTCAAAAGTACCA and ORF-*SsSaxA*-reverse: GGCAGCCGCTGCCTCCGG and cloned into the Gateway cloning vector pDONR™ 201 for sequencing (Invitrogen, Life Technologies Corporation). Subsequently, *SsSaxA* was transferred to the expression vector PH9[45] and the recombinant plasmid was expressed in *E. coli* strain BL 21 (DE 3). The cells were grown until an $OD_{600}$ of 0.5 was reached and thereafter induced with 0.5 mM isopropyl β-D-1-thiogalactopyranoside at 37 °C for 5 h. *E. coli* cells were harvested by centrifugation at $4000 \times g$, 4 °C for 10 min. The pellets were re-suspended in 50 mM Tris-HCl, pH 7.5, 10% (v v⁻¹) glycerol, sonicated (2 × 10% cycle, 60% power, 2 min) and centrifuged at $75,000 \times g$ for 30 min at 4 °C. The crude protein extract was purified with Ni-NTA resin (Qiagen, Hilden, Germany) following the protocols of the manufacturer. Due to non-specific protein binding, a mass western method was used to determine the concentration of SsSaxA protein[46].

Enzyme kinetic studies of SsSaxA were conducted with serial concentrations of 4MSOB-ITC (dissolved in ethanol, 0.1 to 3.0 mM) and 2PE-ITC (dissolved in DMSO, 0.1 to 2.0 mM). Each concentration of substrates was prepared as a 10× stock in solvent. Enzyme assays were performed at 25 °C with the following reaction conditions: 10 μl substrate, 40 mM KP buffer ($KH_2PO_4$-$K_2HPO_4$, pH 7.0), 10 μM $ZnSO_4$ and 0.23 μg recombinant SsSaxA. The final volume of the reaction mixture was diluted to 100 μl with water. After 5 min, the enzyme reaction was terminated with 100 μl methanol. Enzyme reactions of SsSaxA with 8MSOO-ITC were performed following the protocol described above for 4MSOB-ITC and 2PE-ITC using 100 μM of the substrate.

The products of SsSaxA enzyme assays with 2PE-ITC and 8MSOO-ITC were directly analyzed using LC–MS/MS. The amine degradation product of 4MSOB-ITC was first derivatized with FMOC-Cl (9-fluorenylmethoxycarbonyl chloride) due to the signal quenching of the polar 4MSOB-amine by the KP buffer. The FMOC-derivatization of 4MSOB-amine was performed as follows: 100 μl reaction solution was mixed with 100 μl borate buffer (0.8 M, pH 10); 200 μl FMOC-Cl (30 mM in acetonitrile) was added and incubated at room temperature for 5 min; 400 μl hexane was added and mixed to remove unreacted FMOC-Cl. After phase separation, 200 μl of the lower aqueous phase was used for LC–MS/MS analysis. The enzyme kinetic parameters were calculated by SigmaPlot (Systat Software Inc.).

**Effect of 2PE-ITC and its degradation products on fungal growth**. A *S. sclerotiorum* growth assay was conducted with 2PE-ITC and its metabolites 2PE-amine and 2PE-acetamide. These compounds were dissolved in DMSO to generate a series of stock solutions that were added to fungal growth medium (20, 40, 60, 80

and 100 mM). The DMSO stocks were diluted 1:1000 in PDA, and equal amounts of DMSO were added to PDA for the control medium. An agar plug (0.5 cm) with actively growing *S. sclerotiorum* mycelium was placed at the center of the PDA plates amended with the chemical compounds, and the growth diameter of each plate was measured 24 h after inoculation.

**Generation of SsSaxA deletion mutants**. To construct a *SsSaxA* replacement vector, the genomic DNA of *S. sclerotiorum* was isolated with the Stratec Plant DNA Kit (Birkenfeld, Germany) following the manufacturer's protocol. The following primers: 5′flank-*SsSaxA*-forward (*Eco*RI), CGGAATTCTCCGCAATTCAA GTGGCAG and 5′flank-*SsSaxA*-reverse (*Sac*I), CGAGCTCAAGCATATCCCGC CAAGAG; 3′flank-*SsSaxA*-forward (*Sal*I), GTCGACTTGATGGCCGACTTCAA GG and 3′flank-*SsSaxA*-reverse (*Hind*III), AAGCTTGGGGCCATCTTGTTATGT AGC were used to amplify 1174 and 938 bp up- and downstream fragments of the *SsSaxA* genomic region respectively using Phusion taq (New England BioLabs, Boston, MA, USA). The pXEH vector[47] was used as the replacement vector. The flanking regions were cloned into the multiple-cloning sites at both sides of a hygromycin phosphotransferase (hph) cassette by double enzyme digestion and ligation of up- and downstream fragments sequentially using the Quick Ligation Kit (New England Biolabs). Protoplast preparation and PEG-CaCl₂ mediated transformation of *S. sclerotiorum* were performed according to the protocol described by Rollins[48].

The gene replacement was verified by amplification of recombination cassette (Nest) and a 500 bp fragment of the resistance marker gene (*Hyg*) from genomic DNA of WT and mutants with the following primers: Nest-forward: TCAGATT ACCGCTGGGTGC and Nest reverse: TAAAGGTGGGAGGAAAGCAG, *Hyg*-forward: TCCAGAAGAAGATGTTGGCGA and *Hyg*-reverse: GCGAAGAATCTC GTGCTTTCA. Due to the different size of SsSaxA and *Hyg* gene sequences, the size of the recombination cassette is different between WT and gene replacement mutants (1.6 kb for WT and 1.9 kb for mutants). Both WT fungus and transformants that carried the *Hyg* gene and recombination cassette were further induced with 25 μM 2PE-ITC (DMSO as control) to confirm the elimination of *SsSaxA* gene by semi-quantitative PCR of a 549 bp SsSaxA fragment from fungal cDNA with the primers: sq-SsSaxA-forward: CCATCAGGGAGGACTTCGC and sq-SsSaxA-reverse: GGTCCGCTTTAAGCATTCG.

**Phenotypic analysis of ΔSsSaxA mutant**. Both WT and ΔSsSaxA *S. sclerotiorum* were cultured in liquid medium amended with 4MSOB-ITC and samples were collected as described for fungal breakdown product analysis. These strains were also grown on PDA plates containing a series of concentrations of 4MSOB-ITC and 2PE-ITC using the protocols described for the inhibition effect of 2PE-ITC. Fungal pathogenicity was additionally determined by analyzing the lesion area of both WT and ΔSsSaxA on Arabidopsis WT Col-0 using the method described for the assessment of fungal infection levels.

**LC–MS/MS analyses of GLs, ITCs and their fungal metabolites**. Detection of 4MSOB-GL from Arabidopsis was performed on an Agilent 1100 series HPLC system (Agilent Technologies) with a diode array detector. Separation of desulfo-GLs (after on-column desulfation with *Helix pomatia* sulfatase[27]) was achieved on a Nucleodur Sphinx RP C-18 column (250 × 4.6 mm, 5 μm, Macherey-Nagel, Düren, Germany) with water (A) and acetonitrile (B) at a flow rate of 1.0 ml min⁻¹ using a gradient as follows: 1.5% B (1 min), 5% B (5 min), 7% B (2 min), 21% B (10 min), 29% B (5 min), 100% B (1 min), 1.5% B (4 min). The peak area of the LC-UV signal at 229 nm with retention time 8.87 min (desulfo-4MSOB-GL) was used for quantification of 4MSOB-GL with reference to the signal of the internal standard, sinalbin (in the form of desulfo-sinalbin), using a response factor of 2[27].

The analysis of intact 4MSOB-GL, 4MSOB-ITC, 8MSOO-ITC and 2PE-ITC, and their fungal metabolism products were all performed on an Agilent 1200 series HPLC system (Agilent Technologies, Boeblingen, Germany) coupled with an API 3200 tandem mass spectrometer (Applied Biosciences, Darmstadt, Germany).

Separation of intact 4MSOB-GL was achieved on Nucleodur Sphinx RP C-18 column (250 × 4.6 mm, 5 μm) with a solvent system of 0.05% (v v⁻¹) formic acid (A) and acetonitrile (B) at a flow rate of 1.0 ml min⁻¹. The elution profile was the following: 0–1 min, 1.5% B; 1–6 min, 1.5–5% B in A; 6–8 min, 5–7% B in A; 8–11 min, 7–11.2% B in A; 11.1 –12 min, 100% B and 12.1–16 min 1.5% B. The MS parameters were set as follows: ion spray voltage, −4500 V; turbo gas temperature, 700 °C; collision gas, 10 psi; curtain gas, 20 psi; ion source gas 1, 70 psi; ion source gas 2, 60 psi. Parent ion to product ion fragmentation was monitored by multiple reaction monitoring (MRM) in negative ionization mode as follows: $m/z$ 435.9 → 95.8 (collision energy [CE], −60 V; declustering potential [DP], −65 V).

4MSOB-ITC and its fungal metabolism products, including the glutathione conjugate 4MSOB-GSH, 4MSOB-CG, 4MSOB-CYS, 4MSOB-NAC, 4MSOB-amine and 4MSOB-acetamide were separated on a Zorbax Eclipse XDB-C-18 column (50 × 4.6 mm, 1.8 μm; Agilent) with a solvent system of 0.05% (v v⁻¹) formic acid (A) and acetonitrile (B) at a flow rate of 1.1 ml min⁻¹ using a gradient as follows: 0–0.5 min, 3–15% B; 0.5–2.5 min, 15–85% B in A; 2.6–3.5 min, 100% B; 3.5 –3.6 min, 100–3% B in A; 3.6–6 min, 3% B in A. The MS parameters were optimized in positive ionization mode as follows: ion spray voltage, 5500 V; turbo gas temperature, 650 °C; collision gas, 3 psi; curtain gas, 35 psi; ion source gas 1, 60

psi; ion source gas 2, 60 psi. MRM parameters for parent ion to product ion fragmentation were set as follows: $m/z$ 178.11 → 114 (CE, 13 V; DP, 26 V) for 4MSOB-ITC; $m/z$ 485.11 → 179.1 (CE, 29 V; DP, 51 V) for 4MSOB-GSH; $m/z$ 356.07 → 136.1 (CE, 15 V; DP, 21 V) for 4MSOB-CG; $m/z$ 299.06 → 136.1 (CE, 15 V; DP, 26 V) for 4MSOB-CYS; $m/z$ 341.07 → 178.1 (CE, 17 V; DP, 26 V) for 4MSOB-NAC; $m/z$ 136 → 72 (CE, 17 V; DP, 26 V) for 4MSOB-amine; $m/z$ 178 → 114 (CE, 15 V; DP, 26 V) for 4MSOB-acetamide. 8MSOO-ITC and -amine were separated using the same elution profile, and MRM parameters were set as follows: $m/z$ 234 → 170 (CE, 17 V; DP, 36 V) for 8MSOO-ITC; $m/z$ 192 → 128 (CE, 17 V; DP, 26 V) for 8MSOO-amine.

The elution profile for 4MSOB-amine-FMOC was the following: 0–0.5 min, 10% B; 0.5–4.5 min, 10–90% B in A; 4.5–4.51 min, 90–100% B; 4.51–5.0 min, 100% B; 5–5.1 min, 100–10% B and 5.1–8 min 10% B. MRM parameters for 4MSOB-amine-FMOC parent to product ion fragmentation in positive ionization mode were $m/z$ 358 → 179 (CE, 27 V; DP, 46 V).

The elution gradient for 2PE-amine was as follows: 0–4 min, 10–60% B; 4–4.1 min, 60–100% B; 4.1–5 min, 100% B; 5–5.1 min, 100–10% B and 5.1–7 min, 10% B. The MS parameters in positive ionization mode were optimized as follows: ion spray voltage, 5500 V; turbo gas temperature, 700 °C; collision gas, 5 psi; curtain gas, 35 psi; ion source gas 1, 60 psi; ion source gas 2, 60 psi. 2PE-amine fragmentation was monitored by the following MRMs: $m/z$ 122 → 105 (CE, 15 V; DP, 16 V).

The concentrations of these compounds, except 8MSOO-amine and 4MSOB-acetamide, were calculated using an external standard calibration curve generated with the following authentic standards: 4MSOB-ITC, 4MSOB-amine, 4MSOB-GSH, 4MSOB- CYS and 4MSOB- NAC from Santa Cruz Biotechnology (Dallas, TX, USA). 4MSOB-acetamide was quantified relative to the quantity of 4MSOB-amine with an experimental response factor of 1.63[22]. 8MSOO-amine concentrations are reported as relative peak areas. 2PE-ITC and 2PE-amine were purchased from Sigma-Aldrich (Steinheim, Germany). All data were analyzed using Analyst 2.1 (AB-Sciex).

**Statistical analysis**. Data were tested for homogeneity of variances and normality. If data were normally distributed and variances were equal, statistical analysis was performed either with Student's $t$ test, ANOVA or ANCOVA depending on variables and factors using R version 3.1.1, SPSS17.0 and SigmaPlot 12.0. Data were transformed to log or square root if they failed to meet the assumptions of equal variances. The transformed data were then analyzed by corresponding parametric tests. If transformed data still failed to meet the assumption of parametric tests, they were subjected to the non-parametric test followed by a pairwise comparison post-hoc test using SigmaPlot 17.0. The specific statistical method used for each data set is described in the figure legends. Data are presented as mean ± SEM and corresponding $p$ values are indicated in the figure legends.

**Reporting summary**. Further information on research design is available in the Nature Research Reporting Summary linked to this article.

## Data availability

The source data underlying Figs. 1, 2a–d, 3b–c, 4, 5, 6, 7b–c and Supplementary Figs. 1–4, 6–12 are provided as a Source data file.

The accession codes for the SaxA homologs analyzed in this manuscript are: XP_001587011 (*S. sclerotiorum* 1980, SS1G_12040), XP_001587813 (*S. sclerotiorum* 1980, SS1G_11053), XP_001596887 (*S. sclerotiorum* 1980, SS1G_01079), XM_001587066 (*S. sclerotiorum* 1980, SS1G_12145), XP_001584670 (*S. sclerotiorum* 1980, SS1G_14439), ESZ95787 (*S. borealis* F-4128 potential SaxA homolog, putative Zn-dependent hydrolase), ESZ93907 (*S. borealis* F-4128, MBL superfamily protein), ESZ96778 (*S. borealis* F-4128, Putative MBL domain protein), ESZ91064 (*S. borealis* F-4128, SBOR_8558), XP_02454904 (*B. cinerea* B05.10, BCIN_06g00024), XP_024551946 (*B. cinerea* B05.10, BCIN_11g06470), XP_024557191 (*B. cinerea* B05.10, BCIN_02g05980), XP_001552416 (*B. cinerea* B05.10, Bcfmp30), XP_001550649 (*B. cinerea* B05.10, BCIN_05g07490), EGU74976 (*F. oxysporum* Fo5176, FOXB_14513), EGU73472 (*F. oxysporum* Fo5176, FOXB_16024), XP_018158414 (*C. higginsianum* IMI 349063, Zn-dependent beta-lactamase), XP_018153734 (*C. higginsianum* IMI 349063, MBL superfamily protein), WP_011103734 (*P. syringae* DC3000, SaxA), WP_094484734 (*Pectobacterium* sp. *D. radicum*, Drgb3_SaxA). Source data are provided with this paper.

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

## Acknowledgements

This work was financially supported by the Max Planck Society and the China Scholarship Council (CSC, 201406170041). We thank Bettina Raguschke for Sanger sequencing and her assistance in the laboratory, and all members of the greenhouse team, especially Andreas Weber and Elke Goschala, for growing Arabidopsis for this study. The pXEH vector was kindly provided by Prof. Hongyu Pan (Jilin University, Changchun, Jilin).

## Author contributions

J.C., D.G.V., J.G. and A.H. conceived the study. J.C., D.G.V., A.H. and C.U. designed experiments. J.C. performed all the experiments with the help of C.U. and analyzed the data. M.R. and D.G.V. assisted in chemical analysis. F.B. and Z.Y. provided chemical standards and helped with phylogenetic and statistical analyses. A.H. provided additional molecular biology expertise. J.C. prepared draft manuscript with the help from C.U., D.G.V., A.H. and J.G. All authors read, edited and approved the manuscript.

## Competing interests

The authors declare no competing interests.
