## [Peer Review File · Nature Communications]

Reviewers' comments:

Reviewer #1 (Remarks to the Author):

Remarks to the Authors:

The manuscript describes the functional analysis of ITCase (SsSaxA) of the plant pathogenic fungus *Sclerotinia sclerotiorum* (Ss). Showing hydrolytic pathway, previously known only in bacteria, is involved in detoxification of glucosinolate (GS) breakdown products ITCs derived from aliphatic glucosinolates. The authors demonstrated the ability of Ss to hydrolyse 4MSOB-ITC to 4MSOB-amine in liquid culture and the ability of SsSaxA gene to convert toxic ITCs to non-toxic amines in *E. coli* heterologous system. They also demonstrated Δ SsSaxA deficient mutant have reduced ability to convert ITCs to amines in *in-vitro*, reduced resistance to ITCs *in-vitro* and reduced virulence on GS containing *Arabidopsis* plants.

Altogether, the results indicate a link between ITCs hydrolysis and pathogenicity of Ss on Brassicaceae. Although this study is interesting and novel to phytopathogenic fungi I have some major concerns that need to be addressed:

1. In bacteria ITCases have transit peptide. SsSaxA have no TP, as mentioned by authors, nevertheless they measure hydrolysis products in the growth medium. What happens inside the mycelium?
2. Is SsSaxA activity demonstrated is specific to ITCs or general? No specificity to ITCs and no negative controls demonstrated (other compounds such as other ITCs or antibiotics, *E. coli* WT strain)?
3. 4MSOB is indeed abundant in Col-0 but it is also the least toxic ITC, since toxicity dependent on side chain elongation 8MSOB is the most toxic- Is SsSaxA involved in its hydrolysis too? Is chain length effect the activity?
4. Other publications demonstrate enhanced pathogenicity of Ss on both aliphatic and indolic impaired mutants. What about indolic GS and their breakdown products? What about mutants such as *cyp79b2/b3*? Are Δ SsSaxA pathogenicity on such mutants is not altered?
5. Line 113- 114 authors mentioned ITC level reduced in later stages of infection assuming degradation by the enzyme, this is speculative and could be also due to down-regulation of myrosinase activity by the pathogen. Is Myrosinase activity remaining constant during infection? During infection lesion expand and more ITCs should be released.
6. Is 4MSOB-actamide/amine contents correlates with 4MSOB-ITC degradation? (Fig. 2), is it correlates with ITCase activity/ efficiency? Are the sum of ITCs hydrolysis and conjugate products fits the 4MSOB-ITC concentrations?
7. Is SsSaxA expression altered also in high GS levels plants (e.g. MYB overexpressor lines), what happens to its expression levels in Ler ecotype or Col-0 expressing 35S::ESP that contains nitriles and not ITCs?
8. Is expression also upregulated after 2PE-ITC and other ITCs but 4MSOB-ITC exposure?
9. Is there a correlation between growth inhibition on the different concentrations of 2PE-ITC or 4MSOB-ITC and gene expression?
10. Δ SsSaxA pathogenicity of other plant but WT (*tgg 1/2*, *myb28/29*, Col-0:35S::ESP and *cyp79b2/b3*) should be demonstrated to verify effects of ITCs hydrolysis on pathogenicity.
11. Is there more than one KO line demonstrating this effect? or at least a complementation studies should be done.
12. Lesion size on WT in Figure 1 ~0.8 Cm² while in Figure 6 it is almost twice ~2 Cm², is it the same time post inoculation? Time post inoculation is not mentioned in the Fig. 6 legend?

Less important remarks:

1. References in introduction/discussion to be included: Stotz et al., 2011, Rahmanpour et al., 2011, 2014, Pedras&Hossain, 2006, 2011.
2. Are concentration of ITCs used in *in vitro* inhibition assays is not physiologically relevant?
3. Fig2d what concentration of 4MSOB-ITC used?

4. Throughout the manuscript, the figure legends need to be improved, details are missing. E.g. Figs 4, 6F, S4 time after inoculation is missing
5. Is 4MSOB/2PE—amine still not toxic to Δ SsSaxA?

Reviewer #2 (Remarks to the Author):

In this manuscript, the authors present the results of a study where they explored the mechanisms by which the fungal pathogen *Sclerotinia sclerotiorum* tolerates glucosinolate-derived isothiocyanates produced by Brassicales plants. The most interesting and original finding is the description of a new pathway to detoxify ITC in this broad host-range pathogen. Such pathway that relies on an ITC hydrolase has already been described in bacteria but is demonstrated for the first time in the fungal kingdom. In fungi (and in insects), tolerance to ITCs have also been previously reported but was attributed to their capacity to conjugate these molecules to GSH and to export them from the cell. The authors show in this paper that this metabolic pathway also exists in *S. sclerotiorum* but seems far less efficient than the hydrolytic pathway. Interestingly in fungal mutants deficient for ITC hydrolase a compensation mechanism seems to be induced that results in a higher (GSH)-ITC conjugation efficiency. The identified ITC hydrolase is a member of the metallo- β -lactamase-like enzymes that is able to metabolize aliphatic and aromatic ITCs to corresponding non-toxic amines.

Most of the experiments have been carried out correctly and the results are original and of interest for the plant pathology community. Some comments below may help to improve this manuscript:

-Assessment of *S. sclerotiorum* plant colonization (first paragraph of the results section and Fig 1): I am surprised that the authors used the expression of the fungal histone and plant actin genes for normalization to compare the extent of plant colonization by the fungus in the different *At* lines. Why didn't they used direct quantification using genomic DNA as template instead of cDNA? Moreover, the lesions sizes caused by the fungus on leaves are difficult to estimate on the small pictures provided under the graphs.

-Identification of the ITC hydrolase among MBL enzymes in *S. sclerotiorum*. Based on phylogeny and expression data, the authors nicely demonstrate the presence of a specific clade among fungal MBL enzymes that groups with bacterial SaxA proteins (Fig 3). It would be very informative to check whether members of this clade are also present in genomes of *S. sclerotiorum* relatives with narrower host-range and not known as pathogens for Brassicaceae. Such analysis might be difficult for other species within the genus *Sclerotinia* due to the lack of available genomic data but could be easily performed using members of the *Botrytis* genus where genomes of several species with different host-range are available. In line with this, how do the authors interpret the possible absence of SaxA homolog in *S. borealis* that has the ability to infect members of the Brassicaceae family?

-As shown on Fig 6a, the degradation rate of 4MSOB-ITC is lower in Δ SsSaxA than in the WT strain. However, despite this the concentration of this compound significantly decreases over time in the culture medium. Could this be explained only by the higher amounts of ITC-conjugates in the mutant culture? Moreover, it would be interesting to check whether similar results are obtained in planta as this was observed with the WT strain (Fig 2)

-All the pathogenicity assays have been performed on Arabidopsis. It would be informative to compare the aggressiveness of the WT and mutant strains on cultivated Brassica species (e.g. *B. napus*).

-The correct insertion of the disruption cassette in the mutant strain must be checked by Southern blot and/or by PCR to ensure that the replacement construct is inserted at the expected locus (and only at this locus). Moreover, usually to validate the observed mutant phenotype, a control is performed with a complemented strain or alternatively by analyzing more than a single mutant clone.

-Plate assays to determine the dose-dependent effect of ITC and derived compounds on growth of *S. sclerotiorum* WT and mutant strains (legends Fig 5, 6d, 6e): indicate the incubation time for measuring the colony diameter

We would like to thank you for the constructive suggestions on our manuscript and the opportunity to submit a revised version for your consideration. Following your comments, we performed additional experiments that have allowed an extended interpretation of many of our results. With these rigorous amendments, we believe that our manuscript has been greatly strengthened. The changes are described in more detail below, along with a point-by-point response to your comments. We hope, this new version of the manuscript will have addressed all of your concerns and is suitable for publication in Nature Communication.

Reviewers' comments

Responses to comments of Referee #1

Reviewer #1 (Remarks to the Author):

The manuscript describes the functional analysis of ITCase (SsSaxA) of the plant pathogenic fungus *Sclerotinia sclerotiorum* (Ss). Showing hydrolytic pathway, previously known only in bacteria, is involved in detoxification of glucosinolate (GS) breakdown products ITCs derived from aliphatic glucosinolates. The authors demonstrated the ability of Ss to hydrolase 4MSOB-ITC to 4MSOB-amine in liquid culture and the ability of SsSaxA gene to convert toxic ITCs to none toxic amines in *E. coli* heterologous system. They also demonstrated Δ SsSaxA deficient mutant have reduced ability to convert ITCs to amines in – vitro, reduced resistance to ITCs in-vitro and reduced virulence on GS containing *Arabidopsis* plants. Altogether, the results indicate a link between ITCs hydrolysis and pathogenicity of Ss on Brassicaceae. Although this study is interesting and novel to phytopathogenic fungi I have some major concerns that need to be addressed:

We thank the reviewer for their time in evaluating our work and providing some great suggestions. Please see our responses to the specific concerns below.

1. In bacteria ITCases have transit peptide. SsSaxA have no TP, as mentioned by authors, nevertheless they measure hydrolysis products in the growth medium. What happen inside the mycelium?

We thank the reviewers for raising this question. As SsSaxA itself is not secreted and remains intracellular, detoxification is expected to occur inside the fungal cells. ITCs are lipophilic metabolites, and are able to cross membranes and penetrate the mycelial cells. We have now performed additional experiments and chemical analyses of the mycelia of both the wild type fungus and SsSaxA mutants after incubation with ITCs and detected the products of detoxification in these tissues as well. These products are therefore formed intracellularly, and then excreted/transported into the surrounding medium. We have now included these new results in Supplementary Fig. 9.

In addition, we incubated *S. sclerotiorum* *in vitro* for 48 h with 2PE-acetamide, which is the final product of 2PE-ITC degradation. In this assay, levels of 2PE-acetamide did not decline in the medium, suggesting that this metabolite is not taken up by the mycelium, but only transported from the mycelium to the medium.

Figure A: 2PE-acetamide concentrations in the growth medium did not decline during incubation with *S. sclerotiorum* over 48h.

2. Is SsSaxA activity demonstrated is specific to ITCs or general? No specificity to ITCs and no negative controls demonstrated (other compounds such as other ITCs or antibiotics, *E. coli* WT strain)?

Thank you for your suggestion, we have now performed *in vitro* assays of SsSaxA with other potential substrates. SsSaxA was also active towards with 8MSOO-ITC. Since these results are interesting, we have now included them in Supplementary Fig. 5. In addition, we now also show corresponding negative controls in the figures. SsSaxA also hydrolyzed ampicillin *in vitro* (see below), but we did not find any predicted products for other tested lactams (cefotaxime or the synthetic compound vince lactam). However, the biological significance of the hydrolysis of ampicillin (or other lactams) by SsSaxA is unknown and is beyond the scope of our manuscript.

Figure B: Hydrolysis of ampicillin (RT, 2.13 min) by the heterologously expressed SsSaxA protein to ampicilloic acid (RT, 2.02 min). Protein from non-transformed BL21 cells was used as a control. The water-diluted pure substrate also gave rise to degradation products, which were spontaneously formed.

3. 4MSOB is indeed the abundant in Col-0 but it is also the least toxic ITC, since toxicity dependent

on side chain elongation 8MSOO is the most toxic- Is SsSaxA involved in its hydrolysis too? is chain length effect the activity?

SsSaxA can degrade 8MSOO-ITC into the corresponding amine. However, absolute quantification of the product in the assays for a direct comparison is technically impossible at this moment because there is no authentic standard available for this product. However, following the reviewer's comments, we incubated wild-type *S. sclerotiorum* with identical concentrations of 4MSOB-ITC and 8MSOO-ITC over a time course. We found that 8MSOO-ITC was more rapidly degraded by this fungus (Supplementary Fig. 6). Moreover, 2PE-ITC was degraded by wild-type *S. sclerotiorum* at a similar rate as 8MSOO-ITC. We have now added these results and discussed them in the manuscript.

4. Other publications demonstrate enhanced pathogenicity of Ss on both aliphatic and indolic impaired mutants. What about indolic GS and their breakdown products? What about mutants such as *cyp79b2/b3*? are Δ SsSaxA pathogenicity on such mutants is not altered?

We are also convinced that indolic glucosinolate hydrolysis is important for plant defense. However, the *cyp79b2/b3* mutant lacks not only indolic glucosinolates, but also camalexin, a very important phytoalexin active against a range of pathogens including necrotrophs. For this reason, we had initially excluded the analysis of this mutant. However, following the reviewer's suggestion, we have now tested the pathogenicity of Δ SsSaxA mutants and the wild-type fungi on this plant line. Impressively, all three mutants still showed significantly lower virulence on *cyp79b2/b3* plants compared to the wild-type Col-0 plants, highlighting the important role of aliphatic GLs in plant defense. Therefore, we included this information in the last section of the results (Supplementary Fig. 11d). In comparison, wild-type *S. sclerotiorum* was less virulent on Col-0 than on *cyp79b2/b3*, demonstrating the importance of indolic GLs and camalexin in anti-fungal defense.

Figure C: Infection assays using the wild-type *S. sclerotiorum* fungus on Arabidopsis Col-0 and the *cyp79b2/b3* mutant illustrate the importance of indolic GLs/camalexin in anti-fungal defense. Different letters indicate the two groups were statistically different.

5. Line 113- 114 authors mentioned ITC level reduced in later stages of infection assuming degradation by the enzyme, this is speculative and could be also due to down-regulation of myrosinase activity by the pathogen. Is Myrosinase activity remaining constant during infection? during infection lesion expand and more ITCs should be released.

Thank you for pointing this out. We also thought about this scenario at the beginning of our study and performed an experiment to elucidate the role of myrosinase during infection, although we did not include the data in the manuscript. The expression of genes encoding both foliar myrosinases (*tgg1*, *tgg2*) did not change upon fungal infection at 6 and 24 h after inoculation, although a dramatic release of ITCs at 24 h was detected. These results indicated that it is unlikely that the reduction of ITCs during the course of fungal infection was associated with lower myrosinase activity.

Figure D: Transcript abundance of the foliar myrosinases *tgg1* and *tgg2* six and 24 h after fungal infection, compared to mock-inoculated plants.

6. Is 4MSOB-actamide/amine contents correlates with 4MSOB-ITC degradation? (Fig. 2), is it correlates with ITCase activity/ efficiency? are the sum of ITCs hydrolysis and conjugate products fits the 4MOB-ITC concentrations?

The magnitude of ITC disappearance in media is higher than the formation of the amine and ITC-conjugates. To help explain these differences, we have now included chemical analyses of mycelium, where we have detected the accumulation of ITC and different products in the mycelia of both the wild-type and mutant fungi (Supplementary Fig. 9). This indicates that a portion of the ITC being depleted from the medium accumulates intracellularly. A high intracellular accumulation of ITCs (partially as the conjugates) at levels “several hundred-fold over the extracellular concentrations” has been observed in mammalian cells [Zhang, Y., *Carcinogenesis* 22 (2001) 425-31], and was proposed to mediate ITC bioactivities. As this fungus degraded 2PE-ITC more quickly, the quantification of 2PE-amine in the medium at 12 h seems to more closely match the total amount of substrate we supplied in the medium. We are therefore convinced that this pathway is the major metabolism pathway of ITCs in this fungus (Supplementary Fig. 10b).

7. Is *SsSaxA* expression altered also in high GS levels plants (e.g. *MYB* overexpressor lines), what happen to its expression levels in *Ler* ecotype or *Col-0* expressing *35S::ESP* that contains nitriles and not ITCs?

The reviewer raises a good point and we have tried to address the concerns raised by including a larger diversity of plant lines, each containing unique glucosinolate/ITC profiles and concentrations. We performed pathogenicity assays using wild-type *Arabidopsis Col-0*, mutant *Arabidopsis* with altered levels of ITCs (e. g. *tgg1/tgg2*, *myb28/myb29*, *cyp79b2/b3*, and *35S::ESP*) and cultivated Brassicaceae species (*Nasturtium officinale* and *Sinapis alba*). All the assays performed with the different plant lines support the role of *SsSaxA* in overcoming the toxic effects of ITCs when infecting plants of the Brassicaceae.

We also evaluated the induction of *SsSaxA* expression after incubation with 2PE-nitrile in comparison to 2PE-ITC and a solvent control, and observed no induction by the nitrile treatment (Supplementary Fig. 4b). Following the reviewer's comment, we inoculated the 35S::ESP plants using both the wild-type and *SsSaxA* mutant fungi. However, we did not observe a strong reduction of pathogenicity when mutants colonized 35S::ESP plants compared to the wild-type fungus. Therefore, we conclude that *SsSaxA* mutants can colonize nitrile-producing plants about as well as the WT fungus. These results are also included in supplementary Fig. 11c.

8. Is expression also upregulated after 2PE-ITC and other ITCs but 4MSOB-ITC exposure?

Yes, we have now included assays showing that the expression of *SsSaxA* is also upregulated by 8MSOO- and 2PE-ITCs (Supplementary Fig. 4).

9. Is there a correlation between growth inhibition on the different concentrations of 2PE-ITC or 4MSOB-ITC and gene expression?

Expression of the *SsSaxA* gene was significantly induced by different concentrations of 2PE-ITC compared to the control. However, *SsSaxA* expression was higher when the fungus was cultured with lower concentrations (20-40 μ M) of 2PE-ITC than at higher concentrations (up to 100 μ M). There is, therefore, no clear dose-response relationship, probably due to the significant growth-inhibitory/toxic effects at the higher 2PE-ITC concentrations (Supplementary Fig. 4b).

10. Δ SsSaxA pathogenicity of other plant but WT (*tgg 1/2*, *myb28/29*, *Col-0:35S::ESP* and *cyp79b2/b3*) should be demonstrated to verify effects of ITCs hydrolysis on pathogenicity.

Thank you for your suggestion. We have now conducted additional infection experiments on those lines using both the wild-type fungus and Δ *SsSaxA* mutants. These new results have now been included in the manuscript (Supplementary Fig. 11).

11. Is there more than one KO line demonstrating this effect? or at least a complementation studies should be done.

Thank you for pointing this out. Apart from the Δ *SsSaxA*-1 mutant, we have now provided information on two additional knock-out lines (Δ *SsSaxA*-2 and Δ *SsSaxA*-3) throughout this manuscript. These two knock-out lines showed similar phenotypes *in vitro* and *in planta* (Supplementary Fig. 6-9).

12. Lesion size on WT in Figure 1 ~0.8 Cm² while in Figure 6 it is almost twice ~2 Cm², is it the same time post inoculation? Time post inoculation is not mentioned in the Fig. 6 legend?

The times post-inoculation are the same, but the first experiment was performed more than one year before the last inoculation experiment and used different batches of fungal cultures, which can explain the differences between experiments. As such, in this case it is best to compare the results only within experimental blocks, and not among separate experiments. For comparison, we include

data from a separate new inoculation experiment below, which were not included in the manuscript, to provide evidence that the relative infection levels are similar between batches of plants. In this experiment, we have lesion areas of around 1.3-1.5 cm² on WT leaves (Figure E).

Figure E: Lesions created by WT *S. sclerotiorum* on Arabidopsis Col-0 and mutants lacking GL hydrolysis (*tgg1/tgg2*) or aliphatic GLs (*myb28/myb29*)

Less important remarks:

1. References in introduction/discussion to be included: Stotz et al., 2011, Rahmanpour et al., 2011, 2014, Pedras&Hossain , 2006, 2011.

Thank you for the suggestions, we have now included the publications by Stotz et al., 2011 and Pedras & Hossain (2006, 2011). We also added the suggested Rahmanpour et al., 2014 reference, but found that the work by Rahmanpour et al., 2009 fits our discussion better than the paper by Rahmanpour et al., 2011, and included that instead.

2. Are concentration of ITCs used in in vitro inhibition assays is not physiologically relevant?

The concentrations of ITCs used in *in vitro* inhibition assays are roughly those measured in the initial experiments (Figure 2a).

3. Fig2d what concentration of 4MSOB-ITC used?

25 μM 4MSOB-ITC were used in this experiment. This detail is now included in the Methods section.

4. Throughout the manuscript, the figure legends need to be improved, details are missing. E.g. Figs 4, 6F, S4 time after inoculation is missing

Thank you for pointing this out. More details have now been added where we noticed a need for further clarification.

5. Is 4MSOB/2PE—amine still not toxic to ΔSsSaxA?

As ΔSsSaxA mutants do not accumulate amine in the medium and in their mycelia, so we therefore

did not test if these compounds are toxic to mutants, as biologically they will not come into contact with them.

Responses to comments of Referee #2

Reviewer #2 (Remarks to the Author):

In this manuscript, the authors present the results of a study where they explored the mechanisms by which the fungal pathogen *Sclerotinia sclerotiorum* tolerates glucosinolate-derived isothiocyanates produced by Brassicales plants. The most interesting and original finding is the description of a new pathway to detoxify ITC in this broad host-range pathogen. Such pathway that relies on an ITC hydrolase has already been described in bacteria but is demonstrated for the first time in the fungal kingdom. In fungi (and in insects), tolerance to ITCs have also been previously reported but was attributed to their capacity to conjugate these molecules to GSH and to export them from the cell. The authors show in this paper that this metabolic pathway also exists in *S. sclerotiorum* but seems far less efficient than the hydrolytic pathway. Interestingly in fungal mutants deficient for ITC hydrolase a compensation mechanism seems to be induced that results in a higher (GSH)-ITC conjugation efficiency. The identified ITC hydrolase is a member of the metallo- β -lactamase-like enzymes that is able to metabolize aliphatic and aromatic ITCs to corresponding non-toxic amines.

Most of the experiments have been carried out correctly and the results are original and of interest for the plant pathology community. Some comments below may help to improve this manuscript:

We thank the reviewer for their positive evaluation of our work. Please see our responses to the specific comments below.

-Assessment of *S. sclerotiorum* plant colonization (first paragraph of the results section and Fig 1): I am surprised that the authors used the expression of the fungal histone and plant actin genes for normalization to compare the extent of plant colonization by the fungus in the different *At* lines. Why didn't they used direct quantification using genomic DNA as template instead of cDNA? Moreover, the lesions sizes caused by the fungus on leaves are difficult to estimate on the small pictures provided under the graphs.

Thank you for the comment. To determine the pathogenicity on different Arabidopsis lines, we used three independent methods in Fig. 1 and all these methods (phenotypic, lesion area quantification and fungal quantification at the molecular level) were consistent. We increased photo sizes in the figure to make the lesion area more clear. For relative quantification of fungal colonization at the molecular level, we used fungal quantification at transcript levels from cDNA as we found it to be more reliable than at the genomic level, as the mRNA indicates the relative amounts of actively growing fungus present in plant tissue. This technique is used for fungal quantification by a growing number of publications (e.g. doi.org/10.1111/nph.15396; DOI:10.1111/j.1574-6968.2011.02343.x). We have also provided better pictures of the infected plants that were inoculated with either wild-type fungus or Δ SsSaxA mutants (Fig. 7 and Supplementary Fig. 12).

-Identification of the ITC hydrolase among MBL enzymes in *S. sclerotiorum*. Based on phylogeny and expression data, the authors nicely demonstrate the presence of a specific clade among fungal MBL enzymes that groups with bacterial SaxA proteins (Fig 3). It would be very informative to check whether members of this clade are also present in genomes of *S. sclerotiorum* relatives with narrower host-range and not known as pathogens for Brassicaceae. Such analysis might be difficult for other species within the genus *Sclerotinia* due to the lack of available genomic data but could be easily performed using members of the *Botrytis* genus where genomes of several species with different host-range are available. In line with this, how do the authors interpret the possible absence of SaxA homolog in *S. borealis* that has the ability to infect members of the Brassicaceae family?

This is a very interesting question. We searched the available sequence databases rigorously. Among the Sclerotiniaceae, only *S. sclerotiorum*, *Botrytis cinerea* and *S. borealis* have a homolog of the SaxA gene. The amino acid sequence of the *B. cinerea* SaxA candidate gene is included in the phylogeny shown in Figure 3. The *S. borealis* protein is similar to the *S. sclerotiorum* Sax A, with 56% identity and also groups in the SaxA clade together with the bacterial enzymes. However, a closer analysis of the *S. borealis* sequence revealed a premature stop codon at amino acid position 112. We speculate that this might either be a sequencing error, a point mutation specific to the sequenced strain, or that the truncated protein might still have some activity, as the β -lactamase fold is still intact. However, since we currently don't have access to strains from this species, it would be difficult to verify which hypothesis is true. We have thus decided to leave this question open for future research and not include the truncated protein in our phylogenetic analysis.

```

S_bor_put SaxA   MASTFKSDVEI THI GTATAI LSI NGI NMLTDPFFSPAGTQMPTSMEPMLEI TESSAMALH
SS1G_12040      M STFKSTI NI TH I GTATAVLEI DGVNFLTDPFFSPAGSSFPI REDFALEVSEDPALGLN
*  ***** :*****:*. :*:*:*****:..* : : **:*. :. :. :.

S_bor_put SaxA   DLPLV DAVLLSHENHFNDLDDLGRQLLDGRRVLTTPDGAKNLAPRPAVHGL PWESI DTI
SS1G_12040      ELPP I DAVLLSHEDHADNLDLDYGRQLLNGRHVFTTVDGAKNLAPRPAVRGKPVVESTSVN
: ** ***** * ***** *****:*. :* * *****:*. : * * * * .

S_bor_put SaxA   I AGKPKFI TATPCVHFPGHECTGLI VTTPEFG- ETNCLPNAI YFSGDTVYVEELAMRHK
SS1G_12040      LGGVKYTI TATPTQHFPGNECTGFI LTTDRFGHHADGRPNVWFTGDTVYI EEFARI PEQ
: . * : . ***** : * * * * * : * * * * * : * * * * * : * * * * * : * * * * *

S_bor_put SaxA   FHI TI ALFNI GAAW- APEPGVDKMQI TMDGQAARLFRHI EVDI LVPMHFESWKHFTQG
SS1G_12040      FHVVALMNLGSAFVETPI SDGKLVQI TMDGQAGSLFRMLKADHLVPMHYESVGHFTQF
* * * * * : * * * * * : * * * * * : * * * * * : * * * * * : * * * * *

S_bor_put SaxA   KDGLRSAFEAAGI I DHVRVLEPGVAQKI I
SS1G_12040      GKELMADFKEEGVEEKVRVLPVPGAVRI I
. * : * : * : * : * * * * * * * * * *

```

Figure F: Alignment of the *S. sclerotiorum* Sax A protein with the *S. borealis* Sax A protein. The premature stop codon in the *S. borealis* enzyme is highlighted in yellow.

-As shown on Fig 6a, the degradation rate of 4MSOB-ITC is lower in Δ SsSaxA than in the WT strain. However, despite this the concentration of this compound significantly decreases over time in the culture medium. Could this be explained only by the higher amounts of ITC-conjugates in the mutant culture? Moreover, it would be interesting to check whether similar results are obtained in planta as this was observed with the WT strain (Fig 2)

While the concentration of ITC-conjugates produced by the mutant was considerably higher than the WT strain, the extent and speed of ITC disappearance were still higher than the formation of ITC-

conjugates. We have now included chemical analyses of mycelia, where we found a corresponding accumulation of products in WT and mutant mycelia, with more amine products in the WT mycelium but more ITC and conjugates in mutant mycelia (Supplementary Fig. 9). As this fungus degraded 2PE-ITC more quickly, the quantification of 2PE-amine in the medium at 12 h seems to more closely match the total amount of substrate we supplied in the medium, so we are convinced that this pathway is the major metabolism pathway of ITCs in this fungus (Supplementary Fig. 10b).

We have however not performed chemical analyses of plant infection assays with the $\Delta SsSaxA$ mutants, as we predict that the results could be confounded by the natural production of amines by plants under attack (Bednarek et al. (2009), doi: 10.1126/science.1163732 and discussion section).

-All the pathogenicity assays have been performed on Arabidopsis. It would be informative to compare the aggressiveness of the WT and mutant strains on cultivated Brassica species (e.g. B. napus).

We thank the reviewer for this suggestion. We now also performed the pathogenicity assays with WT and mutant fungal strains on watercress (*Nasturtium officinale*) and white mustard (*Sinapis alba*), which are both cultivated and contain benzenic GLs as major GLs. These plants are also significantly more resistant to *SsSaxA* mutants (Supplementary Fig. 12).

-The correct insertion of the disruption cassette in the mutant strain must be checked by Southern blot and/or by PCR to ensure that the replacement construct is inserted at the expected locus (and only at this locus). Moreover, usually to validate the observed the mutant phenotype, a control is performed with a complemented strain or alternatively by analyzing more than a single mutant clone.

Thank you for pointing this out. We have now provided information about two additional knockout lines throughout our manuscript (Supplementary Fig. 7-12). The method of molecular verification of these knockout lines is also included in the supporting information (Supplementary Fig. 7).

-Plate assays to determine the dose-dependent effect of ITC and derived compounds on growth of S. sclerotiorum WT and mutant strains (legends Fig 5, 6d, 6e): indicate the incubation time for measuring the colony diameter

We have now indicated the incubation times in the respective figure legends.

REVIEWERS' COMMENTS:

Reviewer #1 (Remarks to the Author):

The manuscript describes the functional analysis of ITCase (SsSaxA) of the plant pathogenic fungus *Sclerotinia sclerotiorum*. Showing hydrolytic pathway, previously known only in bacteria, is involved in detoxification of glucosinolate (GS) breakdown products ITCs derived from aliphatic glucosinolates.

In the revised version of the manuscript, the authors have addressed all of my previous concerns and comments, added suggested information/data and corrected the MS accordingly.

The presented results convinced me now that the authors have made an important breakthrough with the identification of ITCase as part of the detoxification pathway of plant defense compounds in *S. sclerotiorum*.

I have no further comments or concerns regard the revised MS.

Maggie Levy

Reviewer #2 (Remarks to the Author):

The authors provided satisfactory answers to my comments. However, I think they should provide the reader with information concerning the *S. borealis* homolog of *S. sclerotiorum* SaxA for instance by including this sequence in the phylogeny and discussing in text the fact that this sequence contains a non-sense mutation.

RESPONSE TO REVIEWERS' COMMENTS (2nd round):

Reviewer #1 (Remarks to the Author):

The manuscript describes the functional analysis of ITCase (SsSaxA) of the plant pathogenic fungus Sclerotinia sclerotiorum. Showing hydrolytic pathway, previously known only in bacteria, is involved in detoxification of glucosinolate (GS) breakdown products ITCs derived from aliphatic glucosinolates.

In the revised version of the manuscript, the authors have addressed all of my previous concerns and comments, added suggested information/data and corrected the MS accordingly.

The presented results convinced me now that the authors have made an important breakthrough with the identification of ITCase as part of the detoxification pathway of plant defense compounds in S. sclerotiorum.

I have no further comments or concerns regard the revised MS.

Maggie Levy

Thank you for your support.

Reviewer #2 (Remarks to the Author):

The authors provided satisfactory answers to my comments. However, I think they should provide the reader with information concerning the S. borealis homolog of S. sclerotiorum SaxA for instance by including this sequence in the phylogeny and discussing in text the fact that this sequence contains a non-sense mutation.

Thanks for the suggestion, we now include a sentence referring to this sequence, and how we did not investigate it further.